# An in situ hydrogel-mediated chemo-immunometabolic cancer therapy

Bo Wang [1,4✉], Jing Chen [1,3,4], Julia S. Caserto [2,4], Xi Wang[1] & Minglin Ma [1✉]

Metabolic reprogramming of the tumor microenvironment (TME) and poor immunogenicity are two of the challenges that cancer immunotherapies have to overcome for improved clinical benefits. Among various immunosuppressive metabolites that keep anti-tumor immunity in check, the tryptophan catabolite kynurenine (Kyn) is an attractive target for blockade given its role in mediating immunosuppression through multiple pathways. Here, we present a local chemo-immunometabolic therapy through injection of a supramolecular hydrogel concurrently releasing doxorubicin that induces immunogenic tumor cell death and kynureninase that disrupts Kyn-mediated immunosuppressive pathways in TME. The combination synergically enhances tumor immunogenicity and unleashes anti-tumor immunity. In mouse models of triple negative breast cancer and melanoma, a single low dose peritumoral injection of the therapeutic hydrogel promotes TME transformation toward more immunostimulatory, which leads to enhanced tumor suppression and extended mouse survival. In addition, the systemic anti-tumor surveillance induced by the local treatment exhibits an abscopal effect and prevents tumor relapse post-resection. This versatile approach for local chemo-immunometabolic therapy may serve as a general strategy for enhancing anti-tumor immunity and boosting the efficacy of cancer immunotherapies.

[1] Department of Biological and Environmental Engineering, Cornell University, Ithaca, NY, USA. [2] Robert Frederick Smith School of Chemical and Biomolecular Engineering, Cornell University, Ithaca, NY, USA. [3]Present address: College of pharmacy, Nanjing University of Chinese Medicine, Nanjing 210023, China. [4]These authors contributed equally: Bo Wang, Jing Chen, Julia S. Caserto. ✉email: bowang@cornell.edu; mm826@cornell.edu

Cancer immunotherapy, exemplified by checkpoint blockade, is revolutionizing clinical care for cancer patients[1–3]. By unleashing the intrinsic anti-tumor capabilities of the immune system, immunotherapies can potentially elicit durable responses against cancers[4,5]. Though remarkable in efficacy, a large fraction of patients do not respond to the treatment or develop resistance[6]. Among various factors that contribute to tumor unresponsiveness or relapse, the altered metabolism in the TME that favors cancer cell survival but is hostile to tumor-infiltrating immune cells is being investigated extensively[7–9]. These efforts have led to the identification of multiple metabolites that regulate immune cell functions via different pathways[9]. Theoretically, given the checkpoint-like functions of these metabolites, targeting them offers an alternative means to reinvigorate anti-tumor immunity, independent of immunoinhibitory receptor-blocking antibody administration, therefore potentially expanding the repertoire of general checkpoint blockade-mediated therapeutic modalities and their synergistic combinations. Of these metabolites, tryptophan (Trp) and its catabolite Kyn, generated via the indoleamine 2, 3-dioxygenase (IDO1)/tryptophan 2, 3-dioxygenase (TDO2) pathway, have been under careful scrutiny[10–13]. Mechanistically, both Trp consumption and Kyn production have immunosuppressive effects: the reduction of local Trp concentration resulting from its catabolism activates the general control nonderepressible 2 kinase (GCN2) and suppresses the mechanistic target of rapamycin (mTOR) pathway[14], while the increase of local Kyn concentration activates the aryl hydrocarbon receptor (AHR), which promotes $T_{reg}$ and M2-like macrophage differentiation[15–17]. However, the role of Kyn-mediated immunosuppression in TME is more supported by recent studies[18,19].

Clinically, IDO1 overexpression has been observed in multiple tumor indications, which correlates with CD8$^+$ T cell infiltration and other immunosuppressive pathways[20,21]. Investigations in mouse models suggested that this interferon-γ (IFNγ) driven upregulation of IDO1 serves as negative feedback following CD8$^+$ T cell infiltration[21]. Numerous studies in mouse models have also indicated the role of IDO1 to mediate resistance to programmed cell death protein 1 (PD-1) and cytotoxic T-lymphocyte-associated protein 4 (CTLA-4) blockade and chimeric antigen receptor (CAR)-T treatment[22–24]. These results have inspired the development of small-molecule inhibitors of IDO1, which are currently being tested in clinics[25–27]. However, small-molecule inhibitors are notoriously prone to rapid clearance, induction of drug-resistant mutations, and off-target binding related adverse effects[27–30]. More importantly, these IDO1 inhibitors do not completely prevent Kyn production, as they do not inhibit TDO2[26,27]. On the other hand, enzyme-mediated Kyn elimination (e.g. via kynureninase (KYNase)) has the potential to overcome these limitations, though its efficacy greatly hinges on prolonged catalysis at the tumor site due to similarly fast clearance as small-molecule inhibitors and insufficient accumulation at the target site[31,32]. Therefore, the integration of enzyme-mediated immunotherapeutics with a facile platform capable of sustained and tumor-site local delivery can be a promising solution to resolve these challenges.

Mounting evidence suggests that certain chemotherapies can boost the efficacy of immunotherapy[33–36]. Two primary mechanisms of the synergy between the two distinct therapeutic modalities are induction of immunogenic tumor cell (ICD) of tumor cells and disruption of immunosuppressive components/pathways in TME[33,35,36]. Among multiple chemotherapeutic drugs that are used in clinics, doxorubicin (Dox) is well-known to promote tumor immunogenicity by interfering with DNA synthesis[37,38]. This provides a rational strategy for the combination of Dox and Kyn-degrading enzyme as an effective chemo-immunometabolic therapy: enhancing tumor immunogenicity for T cell targeting and then augmenting T cell activity. However, systemic Dox administration often causes undesired side effects[39,40], e.g., heart damage, and may compromise bone-marrow-resident immune cells to induce immunosuppression[41,42], thus leaving tumor-site local delivery a superior way to synergize with Kyn elimination enzyme-mediated immunotherapeutics.

To overcome the challenges of systemic toxicity of Dox and short half-life of KYNase while leveraging their synergistic therapeutic effects, here we report a localized chemo-immunometabolic therapy that boosts tumor immunogenicity in situ, enhances T cell activity within TME, and induces systemic anti-tumor immunity simultaneously. In this approach, a peritumorally injected supramolecular hydrogel loaded with Dox and Kyn-degrading KYNase releases the drugs locally into the tumor, thereby eliciting ICD, and at the same time, reversing Kyn-mediated immunosuppression (Fig. 1a). The synergy between the two drugs polarizes the TME toward more immunostimulatory, signified by significantly boosted T cell infiltration and activation of TME-residing antigen presenting cells (APCs), in poorly immunogenic 4T1 triple negative breast cancer (TNBC) and B16F10 melanoma mouse models[21,43–45]. TME reprogramming translates into suppression of tumor growth and extended mouse survival. Through analysis of the tumor draining lymph nodes (TdLNs), we confirm that the local combination chemo-immunometabolic therapy induces systemic anti-tumor immunity, which is harnessed to inhibit untreated tumors in the TNBC model. The local combination therapy is also applied as post-operative treatment for prevention of tumor relapse. Therefore, the easy-to-administer and multi-purpose chemo-immunometabolic therapeutic hydrogel offers a facile and robust means to reverse metabolic checkpoint-mediated tumor immunosuppression and may be combined with other checkpoint-blocking therapies for multiple malignant indications as both primary therapy and post-surgical care.

## Results

**IDO1 and TDO2 are upregulated in human TNBC and melanoma.** We first sought to characterize the expression of IDO1/TDO2 in human tumors as a validation for elimination of their product Kyn to boost anti-tumor immunity. Analysis of IDO1/TDO2 expression in the Cancer Genome Atlas (TCGA) database revealed that both IDO1 and TDO2 were upregulated in TNBC and melanoma relative to normal breast and skin tissues (Fig. 1b, c). IDO1 expression strongly correlated with CD3E and CD8A expression in these two tumor types (Fig. 1d, e), indicating that the upregulation was associated with CD8$^+$ T cell infiltration into the tumor, which is consistent with previous reports[21]. Correlation between the expression of TDO2 and that of CD3E and CD8A was also observed, though not as strong as IDO1 (Supplementary Fig. 1). We further investigated the correlation between IDO1 and TDO2 expressions and T cell infiltration markers in other types of human cancer, ranked by their immunogenicity (Supplementary Fig. 2). Interestingly, all tumor types showed strong correlation between IDO1 expression and CD3E and CD8A expressions. TDO2 expression was weakly correlated with CD3E and CD8A expressions. These data suggest that IDO1/TDO2 upregulation is common in TNBC and melanoma, and Kyn may be harnessed as a general metabolic immune checkpoint by different tumors.

To translate these human results into mouse tumor models, we first selected 4T1 breast carcinoma cells, which highly express IDO1[43]. Furthermore, 4T1 tumor is aggressive, highly metastatic, and poorly immunogenic[43], making it a clinically relevant model to evaluate our proposed chemo-immunometabolic therapy for potential human breast cancer treatment. To target Kyn for

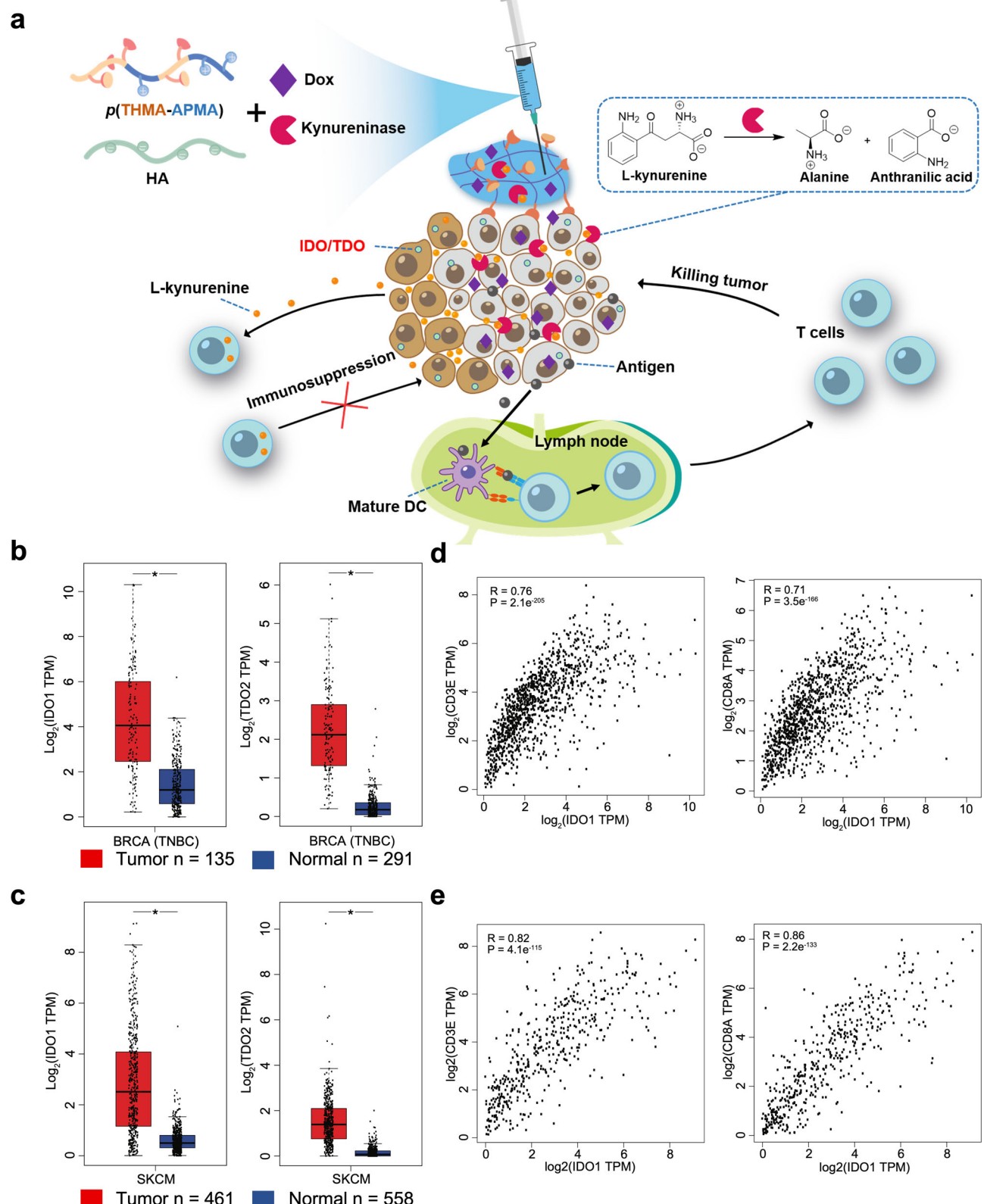

**Fig. 1 The elevated levels of IDO1 and TDO2 and their correlations with T cell infiltration in human TNBC and melanoma support targeting the key metabolite Kyn for tumor suppression. a** Schematics of the localized chemo-immunometabolic therapy enabled by an injectable hydrogel for elicitation of anti-tumor immunity and suppression of tumor growth. **b, c** Expression of IDO1 and TDO2 in TNBC (**b**) and skin cutaneous melanoma (SKCM) (**c**) or the corresponding normal tissues from the TCGA database, represented as transcripts per million (TPM). Whiskers extend to minimum and maximum values, horizontal bars show the median and boxes encompass 25th−75th percentiles, and statistical significance was determined by two-tailed *t*-test. *$P < 0.05$. **d, e** Spearman correlation of IDO1 expression with CD3E and CD8A expressions from the TCGA database for TNBC (**d**, $n = 1085$) and SKCM (**e**, $n = 558$).

augmentation of anti-tumor immunity, we chose *Pseudomonas fluorescens* KYNase that degrades Kyn into alanine and anthranilic acid (Fig. 1a), both of which are nontoxic and immunologically inert (Supplementary Fig. 3a)[31]. The *P. fluorescens* KYNase displayed fast kinetics for Kyn degradation, potentiating sustained elimination of Kyn for unleashing anti-tumor immunity[31].

**The Kyn-degrading immunometabolic hydrogel suppresses 4T1 tumor growth.** To achieve localized tumor therapy, we employed an injectable poly(N-(3-Aminopropyl)methacrylamide)-co-(N-[Tris(hydroxymethyl)methyl]acrylamide) ($p$(AMPA-THMA))-based hydrogel for encapsulation and sustained release of KYNase. The biodegradable hydrogel was designed to be crosslinked through both ionic and supramolecular interactions. We utilized the dynamic interaction between APMA (amine groups) and hyaluronic acid (carboxyl groups) as well as triple hydrogen bonding among THMA to form the injectable hydrogel[46,47]. To adjust the drug release profile, we chose two hydrogel formulations based on different solid contents (10% and 20% by weight, hereafter termed as 10% hydrogel and 20% hydrogel). Angular frequency sweep showed both 10% and 20% hydrogel formulations remained solid-like at all frequencies tested with the storage modulus (G′) remaining above the loss modulus (G″). Meanwhile, G′ was reduced from 7563 Pa for 20% hydrogel to 3706 Pa for 10% hydrogel (Fig. 2a). The yield stress, determined by a stress ramp, showed the same trends (Fig. 2b), achieving approximately 308 and 1074 Pa for 10% and 20% hydrogels, respectively (Fig. 2c). The viscosity of the two hydrogel formulations revealed by a shear rate sweep decreased over 2 orders of magnitude with increasing shear rates, indicating their injectability (Fig. 2d). The shear-dependent viscosity change was also reversible with alternating low ($0.5\,s^{-1}$) and high ($100\,s^{-1}$) shear rates (Fig. 2e, f). Importantly, hydrogel loading did not affect the catalytic activity of KYNase, evidenced by the ability of the encapsulated KYNase to completely degrade Kyn (Supplementary Fig. 3b), despite a delay expected by slower diffusion in the hydrogel.

Having confirmed the injectability of the hydrogel and the bioactivity of encapsulated KYNase, we next examined the in vivo characteristics of the KYNase-loaded hydrogels. KYNase-encapsulated 10% and 20% hydrogels were administered peritumorally in 4T1 tumor-bearing mice, and gel degradation and KYNase release were monitored. We found both 20% and 10% hydrogels displayed an initial increase in gel weight likely due to swelling, followed by gradual degradation (Fig. 2g). By day 10 and 15, the 10% and 20% hydrogels had ~50% mass remaining. On the other hand, 20% hydrogel, with ~80% KYNase released in 15 days, displayed more sustained release of KYNase in comparison with 10% hydrogel that released ~90% KYNase in 10 days (Fig. 2h).

The distinct release profiles of the two hydrogels translated into different metabolite kinetics in TME (Fig. 2i). Both hydrogels showed elimination of Kyn and generation of anthranilic acid 1 day after treatment, with more complete Kyn elimination on day 3. However, the 20% hydrogel enabled more sustained depletion of Kyn (until day 10) in comparison with the 10% hydrogel (until day 5). In addition, the high level of Kyn and the low level of anthranilic acid in PBS-treated tumors confirmed that the elimination of Kyn resulted from the released KYNase. All treatments didn't impact the Trp level significantly.

We explored whether the sustained release of KYNase from hydrogels and depletion of Kyn in TME resulted in anti-tumor effect using the subcutaneous 4T1 TNBC model. When the tumor volumes reached 50 mm³ on day 6, mice were peritumorally treated with KYNase (1.2 mg) loaded in the two hydrogel formulations, the same dose of free KYNase administered once

or every 3 days for a total of three peritumoral injections, or saline (Supplementary Fig. 4). No significant survival benefits were observed for free KYNase treatment, possibly due to the halved dose as previously reported that showed extended survival in CT26 and B16F10 tumor models[31]. However, with this largely reduced KYNase dose, the 20% hydrogel showed substantially suppressed tumor growth and ~28% increase in median survival (36d) relative to three doses of free KYNase (28d). Interestingly, the 10% hydrogel formulation displayed an initial tumor suppression, followed by extensive tumor growth on day 15. This is probably owing to the faster depletion of KYNase from the less crosslinked hydrogel. Hematoxylin and eosin (H&E) staining of the tumor tissues showed that the 20% hydrogel induced significant disruption of tumor microstructure (Supplementary Fig. 5). Both free and hydrogel-loaded KYNase treatments were safe, supported by the consistent body weights of treated mice and no obvious harmful effects on their major organs (Supplementary Fig. 6). We also observed fewer metastatic nodules in the liver of mice treated with the 20% hydrogel (Supplementary Fig. 6b).

To confirm that the therapeutic effect of KYNase-loaded hydrogels resulted from the Kyn-degrading activity of KYNase, but not its immunogenicity, we treated 4T1 tumors with 20% hydrogel loaded with heat-deactivated KYNase (Supplementary Fig. 7). Deactivation of KYNase completely rendered it ineffective in inhibiting the tumor growth, confirming that the therapeutic efficacy of KYNase was not due to its intrinsic immunogenicity.

**The chemo-immunometabolic hydrogel triggers suppression of large 4T1 tumors.** To examine whether it is possible to boost the efficacy of KYNase further, we combined it with the chemotherapy drug Dox. We encapsulated KYNase and Dox using the 20% hydrogel, given its higher efficacy than the 10% one for KYNase single treatment, and applied the combination therapy (1.2 mg KYNase, 60 μg Dox) to even larger 4T1 tumors (~100 mm³) (Fig. 3a). Similar to KYNase release, the hydrogel extended the release of Dox, with ~80% released in 24 h (Supplementary Fig. 8). KYNase formulated in the hydrogel was still able to inhibit tumor growth; however, the addition of Dox significantly enhanced the anti-tumor effect (Fig. 3b, c). On the other hand, peritumoral administration of three doses of KYNase and Dox also augmented tumor suppression and extended mouse survival, but was far less efficacious relative to the hydrogel formulation. Histological analysis revealed extensive disruption of tumor microstructure induced by the combination hydrogel therapy (Supplementary Fig. 9), and no severe adverse effects by the treatments (Supplementary Fig. 10). Liver metastasis was observed to be inhibited by the combo gel (Supplementary Fig. 10b).

To interrogate the changes in TME following different treatments, we performed immunophenotyping of 4T1 tumors 4 days after treatment. Increases in both CD4+ and CD8+ tumor-infiltrating lymphocytes (TILs) were observed post-combination hydrogel treatment, with no significant difference of T_regs in tumors receiving different treatments (Fig. 3d, e, Supplementary Fig. 11f). Interestingly, treatment with the combo gel increased the ratio between CD8+ and CD4+ T cells, indicating that more cytotoxic T cells infiltrated into the tumor. We also observed an increase in the number of CD4+ and CD8+ T cells capable of producing IFNγ, tumor necrosis factor-alpha (TNFα), or interleukin-2 (IL2) (Fig. 3f, g, Supplementary Fig. 11a–d). Furthermore, the combo gel treatment significantly boosted the level of CD39+CD8+ TILs (Fig. 3h, Supplementary Fig. 11e), indicating clonal expansion of tumor-specific T cells after treatment[48]. The upregulation of CD39 also indicates the activation of adenosine-mediated immunosuppressive pathway

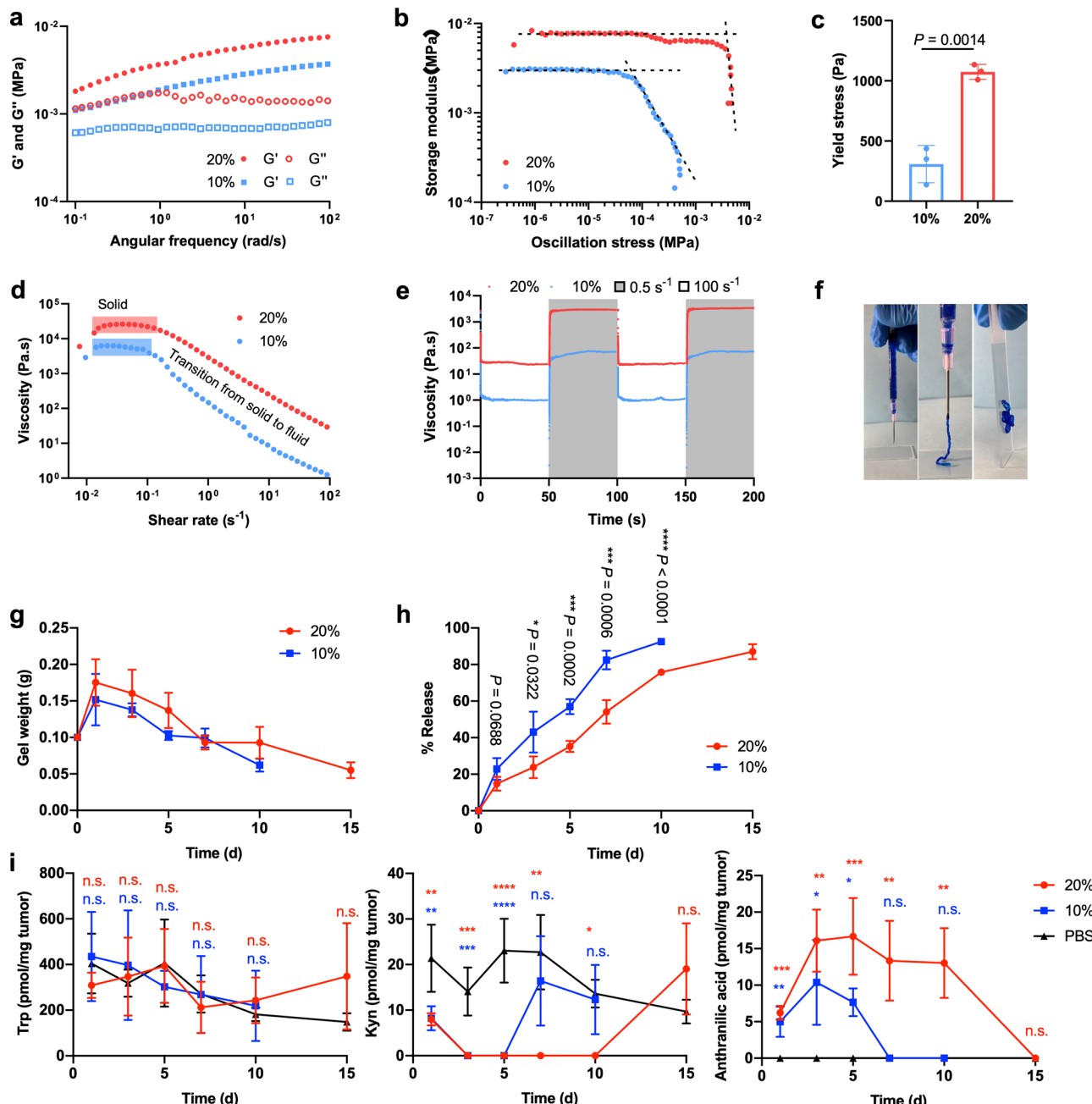

**Fig. 2 The injectable hydrogel empowers sustained KYNase release and Kyn elimination in the tumor. a** Frequency-dependent (strain = 0.1%, 37 °C) oscillatory shear rheology. **b** Stress ramp rheology of two hydrogel formulations. **c** Yield stress values from stress ramp measurements ($n = 3$). **d** Steady shear rheology of 10% and 20% hydrogels. **e** Step-shear measurements of 10% and 20% hydrogels over two cycles with alternating high shear ($100\ s^{-1}$) and low shear ($0.05\ s^{-1}$) rates. **f** Images of 20% hydrogel injection through an 18-gauge needle before injection (left), during injection (middle) and after injection (right) (the hydrogel was mixed with a blue food dye for easier visualization). **g** Degradation kinetics of 10% and 20% hydrogels in 4T1 tumor-bearing mice. **h** KYNase release from 10% and 20% hydrogels in 4T1 tumor-bearing mice over time. **i** Levels of Trp, Kyn, and anthranilic acid in the tumors from 4T1 tumor-bearing mice treated with 10% and 20% hydrogels loaded with KYNase. Metabolite concentrations below the detection limit were assigned to 0. In **g**–**i**, $n = 4$ biologically independent samples for the hydrogel groups, and $n = 3$ for PBS group. Throughout, data are presented as mean ± s.d., and statistical significance was determined by two-tailed $t$-test with Welch's correction (**c**, **h**), or one-way ANOVA with Turkey's post hoc test (**i**, day 1 to day 10) and two-tailed $t$-test with Welch's correction (**i**, day 15). *$P < 0.05$, **$P < 0.01$, ***$P < 0.001$, ****$P < 0.0001$, n.s., not significant. Source data and exact $P$ values are provided as a Source Data file.

after treatment[49]. Together, these data suggest that the combo gel treatment resulted in a more immunostimulatory TME.

Regarding the myeloid lineages, an increase in activated dendritic cells (DCs) and proinflammatory M1-like macrophages was noted after treatment with the combination hydrogel (Fig. 3i, j, Supplementary Fig. 12a, b), while no significant changes of

immunosuppressive M2-like macrophages were identified (Supplementary Fig. 12c). Despite no significant changes of myeloid-derived suppressor cells (MDSCs) between different treatments, this immunosuppressive compartment composed of ~50% of CD11b+ cells in the tumor, posing a challenge for treatment of 4T1 TNBC (Supplementary Fig. 12d).

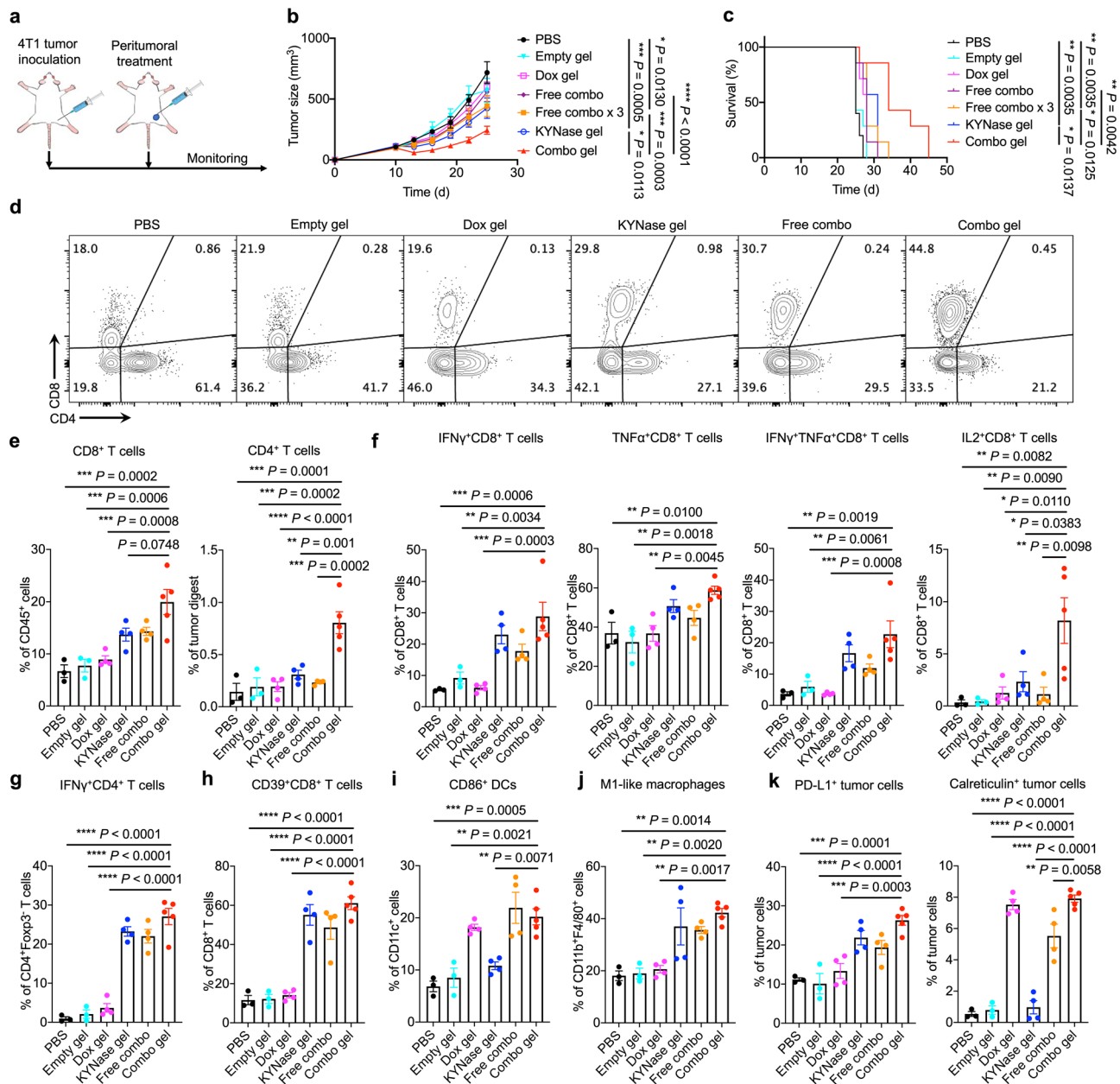

**Fig. 3 Hydrogel-enabled local administration of KYNase and Dox inflames the TME and suppresses established 4T1 TNBC. a** The experimental design. 4T1 cancer cells were implanted subcutaneously on day 0. Mice were treated with empty hydrogel, Dox-loaded hydrogel, KYNase-loaded hydrogel, free KYNase and Dox, free KYNase and Dox administered as 3 doses, or KYNase- and Dox-loaded combination hydrogel on day 10 when tumor volumes reached 100 mm³. **b** Average tumor growth over the course of the study. $n = 5$ mice for PBS and $n = 7$ mice for the other groups. **c** Survival curves for all treatment groups. **d–k** Following 4T1 tumor inoculation on day 0, different treatments were administered on day 10 when tumor volumes reached 100 mm³. Tumors were analyzed 4 days after treatment. $n = 3$ mice for PBS and empty hydrogel groups, $n = 5$ mice for the combination hydrogel group, and $n = 4$ mice for the other groups. **d, e**, Representative FACS plots (**d**) and quantification (**e**) of CD8$^+$ and CD4$^+$ TILs. **f, g** Quantification of IFNγ$^+$, TNFα$^+$, IFNγ$^+$TNFα$^+$, and IL2$^+$ CD8$^+$ (**f**) and IFNγ$^+$CD4$^+$ (**g**) TILs after ex vivo stimulation. **h** Quantification of CD39$^+$CD8$^+$ TILs. **i, j** Percentage of CD86$^+$ DCs (**i**) pre-gated on CD45$^+$CD11c$^+$ populations, and M1-like macrophages (**j**) pre-gated on CD45$^+$CD11b$^+$F4/80$^+$ populations in tumor tissues. **k** Percentage of PD-L1$^+$ and calreticulin$^+$ tumor cells pre-gated on CD45$^-$ populations. Throughout, data are presented as mean ± s.e.m., and statistical significance was determined by two-way ANOVA with Turkey's post hoc test (**b**), one-way ANOVA with Turkey's post hoc test (**e–k**) or log-rank (Mantel-Cox) test (**c**). Source data are provided as a Source Data file.

We assessed whether sustained release of Dox resulted in ICD by profiling calreticulin expression on tumor cells after treatment. Indeed, the combo gel treatment significantly boosted ICD (Fig. 3k, Supplementary Fig. 13b). Moreover, the combo gel treatment also significantly stimulated programmed death-ligand 1 (PD-L1) upregulation on tumor cells, consistent with the enhanced expression potential of IFNγ by CD8$^+$ and CD4$^+$ TILs (Fig. 3k, Supplementary Fig. 13a).

We next evaluated whether local combination therapy elicited systemic anti-tumor immunity by analyzing TdLNs. After restimulation with 4T1 cells, ~6% of CD8$^+$ T cells from TdLNs in mice receiving the combination hydrogel became IFNγ$^+$,

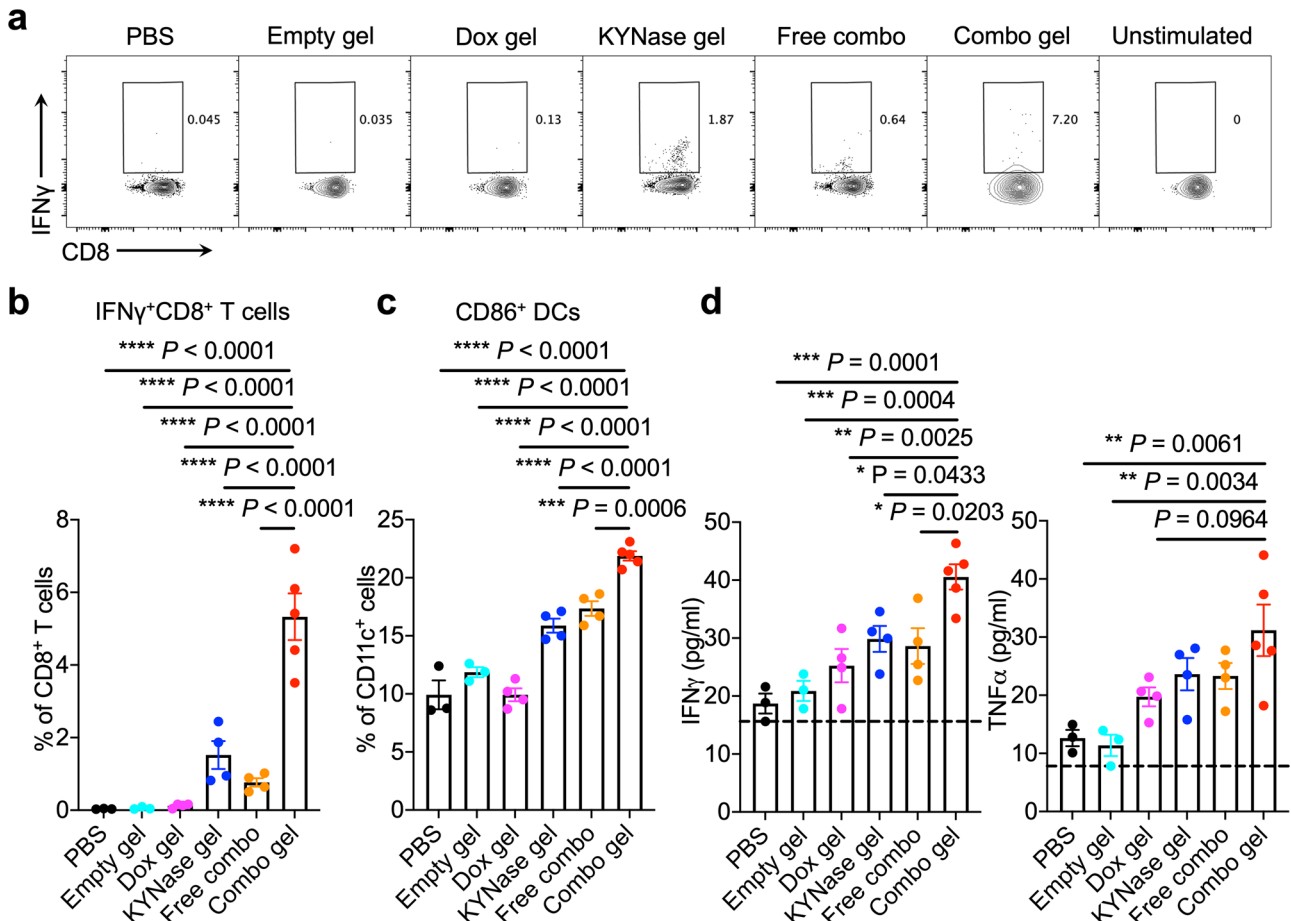

**Fig. 4 The combo gel treatment boosts tumor-specific T cell responses in TdLNs and systemic inflammatory cytokine release.** Following 4T1 tumor inoculation on day 0, different treatments were administered on day 10 when tumor volumes reached 100 mm³. TdLNs were analyzed 4 days after treatment. $n = 3$ mice for PBS and empty hydrogel groups, $n = 5$ mice for the combination hydrogel group, and $n = 4$ mice for the other groups. **a, b** Representative FACS plots (**a**) and percentage (**b**) of IFNγ+CD8+ T cells in TdLNs after restimulation with 4T1 tumor cells. **c** Quantification of CD86+ DCs in TdLNs. **d** Serum IFNγ and TNFα levels 3 days after treatment. The detection limit was annotated with the black line. Data are represented as mean ± s.e.m., and one-way ANOVA with Turkey's post hoc test was used. Source data are provided as a Source Data file.

which was significantly higher than mice receiving other treatments (Fig. 4a, b). CD4+ T cells from TdLNs in the same group of mice also showed increase of IFNγ secretion after 4T1 restimulation (Supplementary Fig. 14a). The mice treated with the combination hydrogel also produced significantly higher number of activated DCs in the TdLNs (Fig. 4c, Supplementary Fig. 14b). As an alternative indicator of systemic anti-tumor immunity, we profiled serum levels of two inflammatory cytokines, IFNγ and TNFα (Fig. 4d). 3 days after different treatments, the combo gel treatment resulted in significant elevation of these cytokines in the serum.

Taken together, these data indicate that the local KYNase- and Dox-mediated chemo-immunometabolic therapy reshaped the TME by eliciting more immunostimulant cells, slowed tumor growth, and induced systemic anti-tumor immunity.

**The combination hydrogel elicits abscopal TNBC suppression and prevents post-operative tumor relapse.** We next sought to explore whether the systemic anti-tumor immunity induced by the local combination chemo-immunometabolic therapy could be harnessed for suppression of untreated tumors. 4T1 tumor cells were subcutaneously inoculated in both flanks of mice, and the tumors on the right flank were treated with combination therapy formulated in the hydrogel or as free, peritumoral doses (Fig. 5a).

Compared with PBS or free dose treatment, significant tumor growth suppression was observed in both the primary tumor site where local combination treatment was administered, and the secondary tumor site that was not treated (Fig. 5b, c). Consistent with the enhanced tumor suppression, immunophenotyping of both tumors from mice receiving different treatments demonstrated that mice treated with KYNase and Dox formulated in the hydrogel elicited significantly more CD8+ and CD4+ TILs (Fig. 5d, e). Furthermore, the combination hydrogel treatment induced more proinflammatory M1-like macrophages in both tumors, with no significant differences between immunosuppressive M2-like macrophages (Fig. 5f, Supplementary Fig. 15a). We also found higher numbers of CD8+ and CD4+ memory T cells in spleens from mice treated with the hydrogel (Supplementary Fig. 15b, c). These results confirmed that the local KYNase and Dox combination therapy can induce systemic anti-tumor immunity.

As another clinically relevant situation, we investigated whether the local combination treatment can inhibit relapse after primary tumor resection, which is relatively common owing to incomplete tumor removal. KYNase and Dox formulated in the hydrogel or as free doses were injected near the original tumor site post-surgical removal of bulk tumors, and tumor re-growth was monitored (Fig. 5g). Strikingly, the hydrogel treatment substantially delayed tumor recurrence, and resulted in ~40%

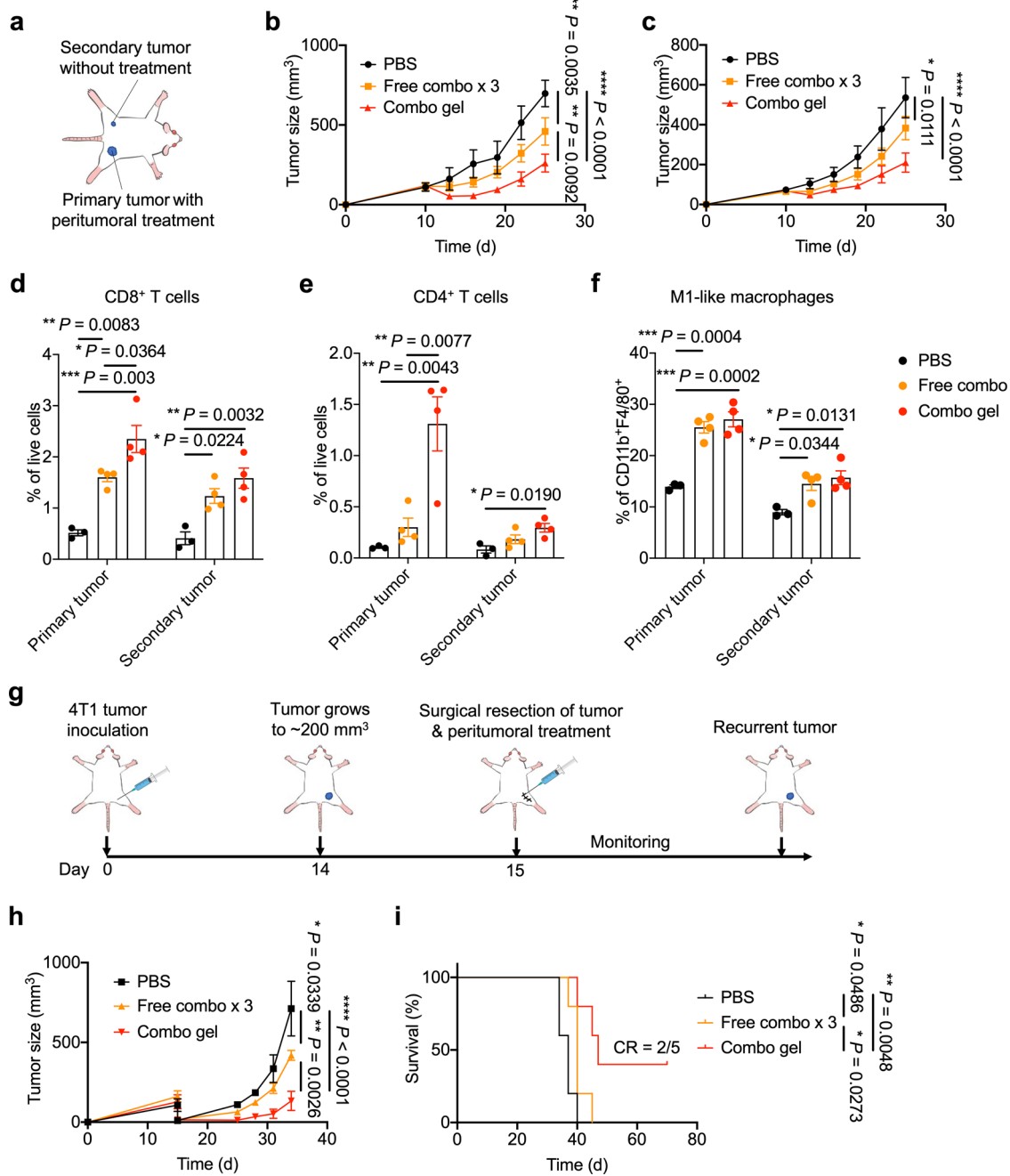

**Fig. 5 Hydrogel-assisted local administration of KYNase and Dox induces abscopal effect and inhibits 4T1 tumor recurrence post-surgery. a** The experimental scheme. 4T1 cells were injected subcutaneously in both flanks on day 0. Mice were treated on right flanks on day 10 with PBS, free KYNase and Dox administered as 3 doses, or KYNase- and Dox-loaded combination hydrogel, when tumor volumes on right flanks reached 100 mm³ (**b–f**). **b, c** Average growth kinetics of primary (**b**) and secondary tumors (**c**), respectively. $n = 5$ mice per group. **d–f** Quantification of CD8⁺ (**d**) and CD4⁺ (**e**) TILs, and M1-like macrophages (**f**) in both primary (treated) and secondary (untreated) tumors that were retrieved on day 25. $n = 3$ mice for PBS and $n = 4$ mice for the other groups. **g** The experimental schedule. Mice were inoculated subcutaneously with 4T1 cells on day 0. On day 15, the bulk tumors were surgically removed and mice were treated with PBS, free KYNase and Dox administered as 3 doses, or KYNase- and Dox-loaded combination hydrogel. **h** Average 4T1 tumor growth over time. $n = 5$ mice per group. **i** Survival for all treatment groups with fraction of complete tumor regression (CR). Data are represented as mean ± s.e.m.. Two-way ANOVA with Turkey's post hoc test (**b, c, h**), one-way ANOVA with Turkey's post hoc test (**d–f**) or log-rank (Mantel-Cox) test (**i**) was conducted. Source data are provided as a Source Data file.

relapse-free survival (Fig. 5h, i). These data indicated that the local KYNase- and Dox-mediated chemo-immunometabolic therapy can effectively inhibit post-operative tumor recurrence.

**The chemo-immunometabolic hydrogel promotes melanoma regression**. To further assess the potential of the hydrogel-

enabled local chemo-immunometabolic therapy, B16F10 melanoma, another aggressive mouse tumor model, was explored. After subcutaneous inoculation, tumors grew to ~85 mm³ before treatment started (Fig. 6a). Mice treated with KYNase and Dox formulated in the hydrogel showed substantial tumor suppression, and ~16% of the mice achieved complete responses (Fig. 6b, c).

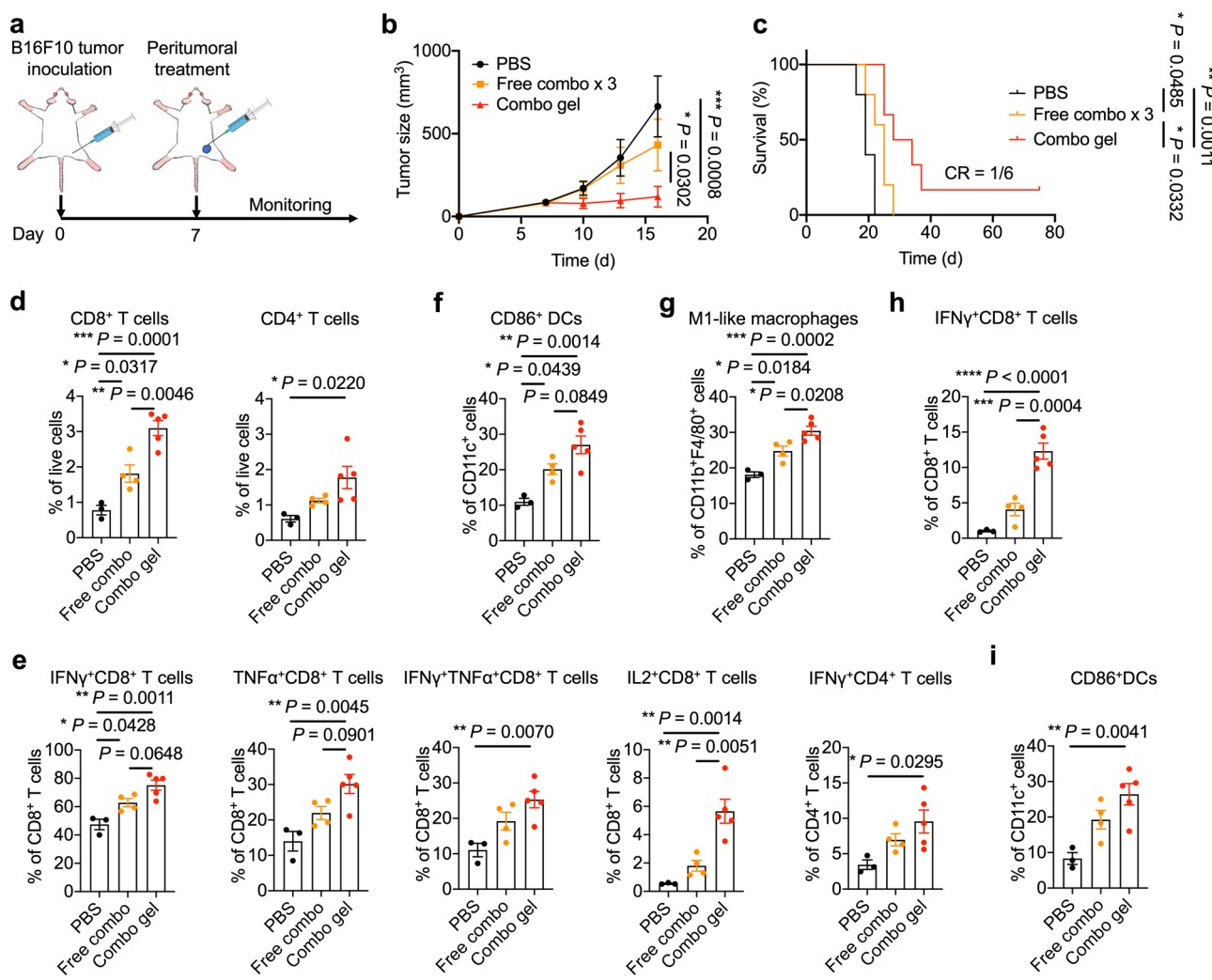

**Fig. 6 Hydrogel-facilitated local administration of KYNase and Dox promotes suppression of B16F10 melanoma. a** The schematics of the experiment. Mice were injected subcutaneously with B16F10 tumor cells on day 0, and treated with PBS, free KYNase and Dox administered as 3 doses, or KYNase- and Dox-loaded combination hydrogel on day 7 when tumor volumes reached 85 mm$^3$. **b** Average tumor growth curves. $n = 5$ mice for PBS and free combination, and $n = 6$ mice for combination hydrogel. **c** Survival analysis for all treatment groups with fraction of complete tumor regression. **d–i** After B16F10 inoculation on day 0, mice were treated on day 7 when tumor volumes reached 100 mm$^3$. Tumors (**d–g**) and TdLNs (**h, i**) were analyzed 4 days post-treatment. $n = 3$ mice for PBS, $n = 4$ mice for free combination group, and $n = 5$ mice for combination hydrogel group. **d** Quantification of CD8$^+$ (left) and CD4$^+$ (right) TILs in tumor tissues. **e** Quantification of IFNγ$^+$, TNFα$^+$, IFNγ$^+$TNFα$^+$, and IL2$^+$ CD8$^+$ TILs and IFNγ$^+$CD4$^+$ TILs after ex vivo stimulation. **f, g** Quantification of CD86$^+$ DCs (**f**) and M1-like macrophages (**g**) in tumor tissues. **h** Percentage of IFNγ$^+$CD8$^+$ T cells in TdLNs after restimulation with B16F10 tumor cells. **i** Quantification of CD86$^+$ DCs in TdLNs. Data are represented as mean ± s.e.m., and two-way ANOVA with Turkey's post hoc test (**b**), one-way ANOVA with Turkey's post hoc test (**d–i**) or log-rank (Mantel-Cox) test (**c**) was used. Source data are provided as a Source Data file.

No significant changes in body weight indicated the safety of the combination therapy in this melanoma tumor model (Supplementary Fig. 16). Immunophenotyping of the tumors from mice receiving different treatments were performed to understand the immunological changes elicited. More CD8$^+$ and CD4$^+$ TILs were noted in mice treated with the hydrogel, similar to the case of 4T1 TNBC, with no significant differences between T$_{reg}$s (Fig. 6d, Supplementary Fig. 17a). Interestingly, we observed significantly more CD8$^+$ TILs capable of producing IFNγ, TNFα, or IL2 (Fig. 6e) in the hydrogel treatment group. The increased number of polyfunctional TILs indicated enhanced TME inflammation and immune activation, therefore improved anti-tumor responses[50]. An increased number of CD4$^+$ TILs capable of secreting IFNγ was also noted (Fig. 6e). Regarding the myeloid lineages, increased numbers of activated DCs and M1-like macrophages were identified, with no significant differences in M2-like macrophages

(Fig. 6f, g, Supplementary Fig. 17b). We also found significant numbers of MDSCs in B16F10 tumors, with no significant difference between mice receiving different treatments (Supplementary Fig. 17c).

We further characterized TdLNs in B16F10 tumor-bearing mice to determine whether a systemic anti-tumor response was induced. After restimulation with B16F10 cancer cells, ~10% of CD8$^+$ T cells in TdLNs from mice treated with the hydrogel secreted IFNγ, significantly higher than those from mice receiving other treatments (Fig. 6h). Modest increase of IFNγ$^+$CD4$^+$ T cells was also noted (Supplementary Fig. 17d). Furthermore, we observed more activated DCs in TdLNs from the hydrogel treated mice (Fig. 6i).

These results demonstrate the broad applicability of the hydrogel-enabled, local administration of KYNase and Dox as an in situ chemo-immunometabolic therapy for treatment of multiple tumors.

## Discussion

In this work, we describe a supramolecular hydrogel-mediated local chemo-immunometabolic therapy composed of Dox for enhancing tumor immunogenicity, and KYNase for unleashing anti-tumor immunity. The majority of current cancer immunotherapeutic strategies depend on the presence of TILs in the TME; however, the integrated TME structures, the hostile immunosuppressive environment, and the poor immunogenicity of some tumors pose significant challenges for improved clinical outcomes[4–8]. On the other hand, the mechanistic identification of anti-tumor immunity enhancement by chemotherapy, namely ICD and disruption of immunosuppression in TME, have resulted in efficacious combinations of chemotherapy and immunotherapy[33,35,36]. Here, we showed that loading of KYNase and Dox into the hydrogel preserved their bioactivity and enabled their sustained local delivery to the tumor upon peritumoral injection. Post administration, the concurrent but differential release of Dox and KYNase first enables ICD and production of tumor antigens, which can be taken up by tumor-residing APCs, evidenced by the increased numbers of proinflammatory myeloid lineages capable of tumor antigen presentation (DCs and macrophages). Subsequently, these activated APCs can travel to TdLNs, where they will stimulate tumor-specific T cells, which will then migrate to the malignant sites to target the tumor cells. On the other hand, the released tumor antigens can be drained directly to TdLNs, where they can be taken up and eventually result in tumor-specific T cell activation. Our analysis of tumor-specific T cells and activated DCs in TdLNs supports such a process of T cell activation. In the tumor sites, these tumor-specific T cells kill tumor cells while overcoming the immunosuppressive microenvironment, including the significant contribution from tumor-derived Kyn. Therefore, sustained Kyn depletion by KYNase released from the hydrogel unleashes Kyn-mediated immunosuppression, supported by the increased numbers of polyfunctional TILs capable of producing IFNγ, TNFα, and IL2. Our data support that the local combination therapy inflamed the TME by stimulating and inducing infiltration of proinflammatory immune cells. Such reshaping of TME toward a more immunostimulatory state boosted anti-tumor immunity, which enhanced tumor growth suppression and prolonged mouse survival in two human cancer relevant models, poorly immunogenic 4T1 TNBC and B16F10 melanoma.

Interestingly, upon treatment, the adenosine pathway (mediated by CD39) was upregulated, and tumor cells also boosted PD-L1 expression. The activation of these alternative immunosuppressive pathways indicates that tumors compensate for the loss of immunosuppressive function caused by KYNase with enhancement of the adenosine and PD1-PD-L1 pathways, which will result in de novo resistance against treatment. Therefore, combinations of the current platform with other therapeutic modalities, e.g., adenosine A2A receptor antagonists or anti-PD-1 antibody, may be considered to further boost the therapeutic efficacy.

Targeting the metabolic checkpoint Kyn with immunotherapeutic KYNase provides an alternative route for unleashing anti-tumor immunity independent of checkpoint-blocking antibody administration. Moreover, the two means of checkpoint blockade, that are, immunoinhibitory receptor-mediated and immunosuppressive metabolite-mediated, may be combined for further efficacy enhancement. It also offers several advantages over small-molecule inhibitors. First, KYNase removes Kyn from the TME regardless of its source, while small-molecule inhibitors only target IDO1 or TDO2 to block Kyn production, leaving the other pathway intact[25,27,51–55]. Second, small-molecule inhibitors are prone to induce drug-resistant mutations in tumor cells[25,27,28]. Third, the structural similarity between small-molecule inhibitors

and AHR agonists may cause some degrees of undesired off-target binding, therefore eliciting some adverse effects[27]. Lastly, the slower release of KYNase relative to small-molecule inhibitors, given its significantly higher molecular weight, enables prolonged Kyn elimination in the TME. As the loaded KYNase is bioactive, the unreleased KYNase in the hydrogel can become a local sink of Kyn considering that Kyn can enter the hydrogel and be degraded there. This potential sink effect of the local combination treatment may further reduce Kyn presence in the TME and enhance anti-tumor responses.

The recent failures of small-molecule IDO1 inhibitors, e.g., epacadostat, in Phase III clinical trials, highlight the significant challenges associated with translating the biology of Kyn pathway into effective cancer therapies[26]. Though initially demonstrating anti-tumor efficacy in a Phase I trial, epacadostat did not show improved efficacy in ECHO-301 trial in comparison with standard care[26]. Multiple reasons have been proposed for the negative outcomes, including lack of patient pre-selection, insufficient IDO1 inhibition, upregulation of IDO2 or TDO2 as compensation for IDO1 blockade, and off-target effect of epacadostat. Interestingly, several recent studies supported the compensation hypothesis, rendering dual blockade of IDO1 and TDO2 essential[56,57]. As described above, KYNase will have no off-target effect and remove Kyn regardless of its source, thus may be a more appropriate candidate for unleashing Kyn-mediated immunosuppressive pathway in clinics.

The remodeling of TME by the local combination therapy elicited a systemic anti-tumor response, supported by the increased numbers of tumor cell-specific T cells in TdLNs capable of producing IFNγ upon re-encounter with cancer cells. Utilizing this systemic response, the local combination therapy could induce abscopal effect in untreated 4T1 tumors.

Besides primary tumor treatment, the local combination therapy was used for prevention of tumor relapse after resection. Surgical removal of tumor tissues is an indispensable treatment strategy for many cancers. However, the incomplete removal of tumor cells in the primary site, together with the inoperable metastatic sites, often results in tumor reoccurrence[58]. Our local chemo-immunometabolic therapy can take advantage of the residual tumor to elicit the immune response through Dox-mediated ICD, and then boost the response through KYNase-mediated reversal of immunosuppression. The systemic anti-tumor responses elicited by the local treatment, as demonstrated, can target the metastatic sites. Therefore, the combination therapeutic hydrogel may offer superior post-operative care.

In conclusion, our local chemo-immunometabolic therapy offers a flexible strategy for tumor combination therapy by simultaneously targeting tumor cells cytotoxically and metabolically. Our findings suggest that this local combination therapy has the potential to sensitize tumors to immunotherapy and enhance anti-tumor immunity in a safe and effective manner.

## Methods

**Animals and cell lines**. Male and female C57BL/6 and female BALB/c mice were purchased from the Jackson Laboratory and were 6–10 weeks old at the beginning of each experiment. The mice were housed on a 12 h light-dark cycle at 20–26 °C with 30–70% humidity. All animal studies were performed following the protocols approved by Cornell University's Institutional Animal Care and Use Committee and in compliance with all relevant ethical regulations.

4T1 breast cancer and B16F10 melanoma cell lines were obtained from X. Wang and C. Kenific (Lyden lab) at Cornell University. They are cultured in RPMI 1640 (Gibco) supplemented with 10% HI-FBS (R&D Systems) and DMEM (Gibco) supplemented with 10% HI-FBS, respectively.

**Chemicals**. Food grade sodium hyaluronate (hyaluronic acid) powder of 1.8MD molecular weight was purchased from Prescribed for Life. N-(3-Aminopropyl) methacrylamide Hydrochloride (APMA) was purchased from Accela. Sodium dodecyl sulfate (SDS), Picrylsulfonic acid solution (TNBS, 5% (w/v) in $H_2O$),

Ammonium persulfate (APS), N-[Tris(hydroxymethyl)methyl]acrylamide (THMA), acetone, and HCl were purchased from Sigma-Aldrich. Dox was purchased from Sigma-Aldrich.

**Synthesis of _p_(APMA-THMA)**. _p_(APMA-THMA) was synthesized and purified by as previously reported[46,47]. APS (23 mg, 0.1 mmol), THMA (728 mg, 4.16 mmol), and APMA (185 mg, 1.04 mmol) were dissolved in 6 ml of deionized water by stirring with nitrogen bubbling for 30 min at room temperature. The mixture was stirred at 60 °C for 1 h and then at 70 °C for an additional 12 h. _p_(APMA-THMA) was precipitated by dropping final reaction mixture into 15-20-fold volume of acetone, and then redissolved in water and dialyzed against water (Spectra/Pore, 3.5kD molecular weight cut off (MWCO)) for 4 days. The final polymer powder was obtained by lyophilization. The degree of substitution (DS) of APMA in _p_(APMA-THMA) was characterized using TNBS assay according to a previously reported method[46,47]. The DS of APMA was determined to be 15.7%.

**Supramolecular hydrogel preparation**. 10% and 20% (wt/wt) solid contents of supramolecular hydrogels were prepared by mixing pre-determined amounts of hyaluronic acid and _p_(APMA-THMA) corresponding to a 1:1 molar ratio of primary amines to carboxyl groups. The solutions were sufficiently mixed with a spatula, mildly centrifuged to remove bubbles from mixing before loading into a syringe.

**Preparation of KYNase**. The _P. fluorescens_ KYNase was prepared as described previously[31]. Briefly, the pET28a vector encoding the _E. coli_ codon-optimized and 6x histidine-tagged KYNase was transformed into _E. coli_ BL21(DE3) strain. 8 ml starter cultures of LB (BD Difco) supplemented with 50 μg/ml kanamycin (Sigma) were inoculated with single colonies and cultured at 37 °C overnight (o/n). 4 L TB (BD Difco) plus 50 μg/ml kanamycin were inoculated with the starter culture at 1:500 dilution the next day, and cultured at 37 °C until OD$_{600}$ of 0.8. 0.5 mM IPTG (Sigma) was added to induce KYNase expression at 18 °C o/n. Cells were then spun down and resuspended in lysis buffer containing 50 mM sodium phosphate, pH 8.0, 1 mM pyridoxal-5-phophate (PLP), 300 mM NaCl, 25 mM imidazole, and 0.1% Tween-20. Cells were lysed by sonication, and cleared by centrifugation at 20,000 g for 30 min at 4 °C. The supernatant was filtered and loaded onto a customized column prepacked with 8 ml HisPur Ni-NTA (Thermo Scientific) at 4 °C. The column was washed with 100 ml lysis buffer, followed by 2 L endotoxin-free PBS (Corning), pH 7.4, supplemented with 0.2% Triton X-114 (Sigma). After another wash with 100 ml PBS, KYNase was eluted with endotoxin-free and filtered elution buffer containing 50 mM sodium phosphate, pH 8.0, 1 mM PLP, 300 mM NaCl, and 250 mM imidazole. KYNase was then buffer exchanged into PBS and concentrated with Amicon Ultra centrifugal filters (30kD MWCO, Millipore Sigma). KYNase was sterile-filtered, flash-frozen in liquid nitrogen, and stored at −80 °C until use. After at least 24 h, a small aliquot of KYNase was thawed and used to check catalytic activity, endotoxin contamination, and integrity. Catalytic activity was determined by monitoring absorbance at 365 nm over time for Kyn degradation. Endotoxin levels were determined by Pierce Chromogenic Endotoxin Quant Kit (Thermo Scientific), and were <0.1EU/mg. KYNase integrity was checked by SDS-PAGE on a 4-20% Mini-PROTEAN TGX gel (BioRad), and were >98% pure.

**Rheological testing**. Frequency sweeps over an angular frequency range of 1 to 100 rad/s were conducted with a strain of 0.1%. Yield stress values were obtained from stress ramp measurements (n = 3). Flow ramp measurements with a shear rate range of 0.01–100 s$^{-1}$ were conducted to observe the effects of viscosity versus shear rate. Step-shear measurements were taken by alternating high shear (100 s$^{-1}$) and low shear (0.5 s$^{-1}$). Each step was 50 seconds. All measurements were taken at 37 °C utilizing a 20 mm aluminum Peltier plate to simulate rheological behavior in vivo. All data was analyzed using TA Instruments Trios software and plotted using GraphPad Prism 9.

**Enzymatic analysis**. 40 μl of 5 μM KYNase, either as free enzyme solution, or encapsulated in 50 μl of 20% or 10% hydrogels, was added into a 96 well plate. Reactions were initiated by adding 160 μl of 1.25 mM Kyn to the enzymes. Absorbance at 365 nm was monitored with a plate reader pre-heated to 37 °C, and the time needed for complete elimination of Kyn in the wells was analyzed for each formulation.

**Hydrogel degradation and KYNase and Dox release**. Tumors were established by injecting 10$^6$ 4T1 cancer cells subcutaneously into the right flank of female BALB/c mice. After 6 days when tumor volumes reached 50 mm$^3$, mice were treated peritumorally with 60 mg/kg KYNase in 10% hydrogel, 60 mg/kg KYNase in 20% hydrogel, or 3 mg/kg Dox in 20% hydrogel. The KYNase loaded into the hydrogels were pre-labeled using Cy5 NHS ester (Abcam). At 1, 6, 12, 24, 72, and 120 h after Dox gel injection, mice were euthanized, and the hydrogels were harvested. At 1, 3, 5, 7, 10, and 15d after KYNase gel administration, mice were sacrificed, and the hydrogels were collected and weighed.

500 μl PBS was added to the retrieved hydrogels, which were then homogenized to release the residual Dox and KYNase in them. After homogenization, the samples were centrifuged at 5000 g for 10 min, and 100 μl supernatant were used for fluorescence analysis with a plate reader. The fluorescence of Dox was monitored at 595 nm emission with 470 nm excitation, and the fluorescence of Cy5-labeled KYNase was monitored at 666 nm emission with 633 nm excitation.

**Metabolites analysis**. Tumors were established as described above. After 6 days when tumor volumes reached 50 mm$^3$, mice were treated peritumorally with 60 mg/kg KYNase in 10% hydrogel, 60 mg/kg KYNase in 20% hydrogel, or PBS. At 1, 3, 5, 7, 10, and 15d after treatment, mice were sacrificed, and the tumors were collected and weighed.

Tumor homogenization was performed similarly as previously described[31]. Briefly, 500 μl of a chilled 50:50 water:methanol (v:v) solution was added to tumor samples, after which they were homogenized. The homogenate was extracted with 500 μl of chilled chloroform, and then centrifuged at 5000 g for 20 min at 4 °C. The aqueous phase was harvested, desiccated, and resuspended in water. HPLC quantification was performed on a C-18 reverse phase column with a mobile phase of methanol:water (v:v, 30:70) at a flow rate of 0.8 ml/min. As described[32], Trp was quantified at 270 nm absorption, Kyn was quantified at 360 nm absorption, and anthranilic acd was quantified at 330 nm absorption.

**Tumor models and therapy**. For single 4T1 tumor treatment, 10$^6$ 4T1 cancer cells were injected subcutaneously into the right flank of female BALB/c mice. After 6 days (treatment with KYNase) or 10 days (treatment with KYNase and Dox) when tumor volumes reached 50 mm$^3$ or 100 mm$^3$, respectively, mice were treated peritumorally with PBS, 60 mg/kg KYNase, 3 doses of 20 mg/kg KYNase administered every 3 days, 60 mg/kg KYNase in 10% hydrogel, or 60 mg/kg KYNase in 20% hydrogel for 50 mm$^3$ tumors, or PBS, 20% hydrogel, 3 mg/kg Dox in 20% hydrogel, 60 mg/kg KYNase and 3 mg/kg Dox, 3 doses of 20 mg/kg KYNase and 1 mg/kg Dox administered every 3 days, 60 mg/kg KYNase in 20% hydrogel, or 60 mg/kg KYNase and 3 mg/kg Dox in 20% hydrogel for 100 mm$^3$ tumors. Tumor sizes and body weights were tracked 2-3 times weekly. Tumor volume was calculated using length x width$^2$/2. Mice were euthanized when tumor volumes reached 1,000 mm$^3$ or 500 mm$^3$ with ulceration.

For abscopal effect against 4T1 tumors, 10$^6$ and 5×10$^5$ 4T1 cells were injected subcutaneously into the right and left flank of BALB/c mice, respectively. After 10 days when tumor volumes reached 100 mm$^3$ in the right flank, mice were treated peritumorally in the right flank with PBS, 3 doses of 20 mg/kg KYNase and 1 mg/kg Dox administered every 3 days, or 60 mg/kg KYNase and 3 mg/kg Dox in 20% hydrogel. Tumor size was monitored as above. Mice were euthanized on day 25 and tumors and spleens were harvested for analysis.

For 4T1 tumor relapse, 10$^6$ 4T1 cells were injected subcutaneously into the right flank of BALB/c mice. After 15 days, a small incision ~1 cm was created on the tumor site by a sterile scalpel, and the bulk tumor was removed, leaving ~5% residue. After the incision was sutured, PBS, 20 mg/kg KYNase and 1 mg/kg Dox, or 60 mg/kg KYNase and 3 mg/kg Dox in 20% hydrogel was injected into the surgical site. 2 more doses of 20 mg/kg KYNase and 1 mg/kg Dox was administered in mice receiving this treatment 3 and 6 days later. Tumor size and mouse survival were tracked as above.

For B16F10 tumor treatment, 10$^6$ B16F10 melanoma cells were injected subcutaneously into the right flank of C57BL/6 mice. After 7 days when tumor volumes reached 85mm$^3$, mice were treated peritumorally with PBS, 3 doses of 20 mg/kg KYNase and 1 mg/kg Dox administered every 3 days, or 60 mg/kg KYNase and 3 mg/kg Dox in 20% hydrogel. Tumor size and mouse survival were monitored as above.

**Immunophenotyping of tumors, TdLNs, and spleens**. Tumors were established and treated as described above. At indicated time points, mice were sacrificed, and the respective tissues were collected. Tumors were cut into small pieces and digested in dissociation buffer (RPMI 1640, 1 mg/ml collagenase IV (Worthington Biochemical Corporation), and 100 μg/ml DNAse I (Thermo Scientific)) at 37 °C for 45 min with gentle shaking. The cell suspension was passed through a 70μm cell strainer and treated with ACK lysing buffer (Gibco). TdLNs were mechanically digested and passed through a 70μm cell strainer to obtain single cell suspensions. Spleens were mechanically disrupted, filtered through a 70μm cell strainer, and treated with ACK lysing buffer to obtain single cell suspensions.

Single cell suspensions were stained with Ghost Dye UV450 (Tonbo Biosciences) at 4 °C for 20 min, then blocked with TruStain FcX Plus (BioLegend) for 10 min. After washing, they were stained with antibodies against surface proteins at 4 °C for 30 min. To detect cytokine production, single cell suspensions of tumor tissues were resuspended in T cell medium (RPMI 1640, 10% HI-FBS, 50 μM 2-β-mercaptoethanol (Gibco), and 1x Penicillin-Streptomycin (Gibco)) and stimulated with 1x cell activation cocktail (BioLegend) at 37 °C for 1 h, after which Brefeldin A (Biolegend) was added. Cells were cultured for another 4 h, and then processed as described above for surface markers. The intracellular staining was then performed using eBioscience Foxp3/Transcription Factor staining buffer set. Single cell suspensions of TdLNs were resuspended in T cell medium and cocultured with 4T1 cells or B16F10 cells for 12 h, after which Brefeldin A was

added. After incubation for additional 6 h, cells were processed for surface marker staining and then intracellular staining using BD Cytofix/Cytoperm Fixation/ Permeabilization kit. A BD FACSymphony A3 was used for flow cytometric data acquisition, and data were collected and analyzed with BD FACSDiva 8.0.1 and Flowjo 10.7.1, repsectively.

The following antibodies were used: CD45-BV650 (BioLegend 103151, 30-F11, 1:200), CD3-Pacific Blue (BioLegend 100213, 17A2, 1:200), PD-L1-PE (BioLegend 124307, 10 F.9G2, 1:100), Calreticulin-Alexa Fluor 647(R&D Systems IC38981R-100UG, 1:100), CD62L-FITC (BioLegend 104405, MEL-14, 1:200), CD8-PE (BioLegend 100707, 53-6.7, 1:200), CD8-APC-Cy7 (BioLegend 100713, 53-6.7, 1:200), CD8-Alexa Fluor 700 (BioLegend 100729, 53-6.7, 1:200), CD44-APC (BioLegend 103011, IM7, 1:200), CD4-APC-Cy7 (Tonbo Biosciences 25-0041-U025, GK1.5, 1:200), CD4-BV785 (BioLegend 100551, RM4-5, 1:200), CD4-AlexaFluor 594 (BioLegend 100446, GK1.5, 1:200), IL2-FITC (BioLegend 503805, JES6-5H4, 1:100), TNFα-PE (BioLegend 506305, MP6-XT22, 1:100), Foxp3-eFluor615 (eBioscience 42-5773-82, FJK-16s, 1:100), CD39-PE-Cy7 (BioLegend 143805, Duha59, 1:200), IFNγ-APC (BioLegend 505809, XMG1.2, 1:100), CD11c-FITC (BioLegend 117305, N418, 1:200), CD11c-APC (BioLegend 117309, N418, 1:200), F4/80-PE (BioLegend 123109, BM8, 1:200), CD86-PE-Cy7 (BioLegend 105013, GL-1, 1:200), CD11b-APC-Cy7 (BioLegend 101225, M1/70, 1:200), Gr-1-FITC (Tonbo Biosciences 35-5931-U025, RB6-8C5, 1:200), and CD206-PE-Dazzle594 (BioLegend 141731, C068C2, 1:200).

**Detection of cytokines in serum**. 3 days after different treatments, the blood samples were collected and coagulated at room temperature for 30 min. Serum was then collected after centrifugation at 15,000 g for 15 min. IFNγ and TNFα were detected using ELISA MAX Deluxe Set Mouse IFNγ and TNFα (BioLegend), respectively.

**Histology**. Different organs, including heart, liver, spleen, lungs, kidneys, and tumors, were harvested and fixed in formalin. They were subsequently embedded in paraffin, sectioned and stained with hematoxylin and eosin. Images were taken under a microscope.

**Analysis of TCGA data**. IDO1 and TDO2 expression in human TNBC and melanoma versus counterpart normal tissues were analyzed using TCGA databases, and the mean values were calculated. IDO1 and TDO2 correlation with CD3E and CD8A for human cancers were obtained from TCGA and analyzed by GEPIA2.

**Statistical analysis**. Statistical significance was determined by two-tailed $t$-test with Welch's correction for comparison of two groups, one-way or two-way ANOVA with Turkey's post hoc test for comparison of multiple groups or log-rank (Mantel-Cox) test for survival analysis. Samples sizes are indicated in figure legends. All statistical analyses were performed with GraphPad Prism 9.

**Reporting summary**. Further information on research design is available in the Nature Research Reporting Summary linked to this article.

## Data availability

Human tumor data were obtained from TCGA databases [https://portal.gdc.cancer.gov/]. All data generated from this study are available within the Article, Supplementary Information or Source Data file. Source data are provided with this paper.

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

## Acknowledgements

We thank C. Kenific (Lyden lab) at Cornell University for providing B16F10 cell line, Q. Liu and D. Gao (Ma lab) at Cornell University for helpful discussions on hydrogels, Cornell Histology Core Facility for the histological sectioning and staining, Cornell Flow Cytometry Core Facility for the use of BD FACSymphony A3, Cornell Department of Human Ecology for the use of HPLC, and Cornell Center for Materials Research Facility for the hydrogel characterization, which is supported by the NSF under award number DMR-1719875. This work was supported by the Hartwell Foundation (M.M.) and Cornell COVID-19 seed grant (B.W.).

## Author contributions

B.W., J.C., and M.M. designed the study, B.W., J.C., and J.S.C. performed experiments, B.W., J.C., and J.S.C. analyzed and interpreted the data, B.W., J.C., J.S.C. and M.M. wrote the manuscript, X.W. provided reagents.

## Competing interests

The authors declare no competing interests.
