## [Peer Review File · Nature Communications]

An in situ hydrogel-mediated chemo-immunometabolic cancer therapyREVIEWER COMMENTS

Reviewer #1 (Remarks to the Author): with expertise in nanomedicine

In this manuscript, the authors reported a chemo-immunometabolic strategy using a supramolecular hydrogel that releases doxorubicin (Dox) and kynureninase (KYNase) to induce immunogenic tumor cell death and disrupt Kyn-mediated immunosuppressive pathways for enhanced cancer treatment. The authors demonstrated the improved tumor inhibition and extended survival of such hydrogel in both triple negative breast cancer and melanoma. The design has a high translation potential and the data is promising. I recommend its publication after addressing the following technical issues.

1. The authors demonstrated the positive correlation between IDO1/TDO2 expression and CD3E and CD8A expression in TNBC. How does this correlation vary with the tumor immunogenicity?
2. The release profiles of Dox and KYNase from the hydrogel should be provided.
3. According to Figure 3d and 3c, the tumor inhibition and survival improvement of the combo gel are not very effective, with the tumor rapidly growing after 20 days. Why does the treatment of Dox and KYNase (even three times) show subtle tumor inhibition effect and similar survival with the PBS group? It is noted that the drug was intratumorally injected, which should have high tumor-killing effect.
4. In Figure 3d, there are some issues regarding the flow cytometric plots of CD4 and CD8 cells.
 - a. The boundary of cell grouping is not very clear. There are multiple cell populations within the CD4+ gate (including PBS, Free combo, and combo gel groups). Moreover, a lot of cells were excluded from the CD8+ gate but seem the same population with those in the gate.
 - b. A group of CD4+CD8+ cells is obvious in KYNase gel and Combo gel groups, which is uncommon in flow cytometric analysis. Some explanations should be provided here.
 - c. It is difficult to distinguish CD4-CD8- cells in most of the groups.
 - d. In Dox gel group, there are many cells at the left bottom corner, which seems to be cropped from the original image.
 - e. The percentage labels in all flow cytometric plots cover the cell gating, which greatly affect data presentation.
5. Some issues of other flow cytometric analysis.
 - a. It seems that there is only one cell population in the plots of Treg analysis in Supplementary Fig.9a and CD8+IFN- γ + cell analysis in Figure 3i and Supplementary Fig.9d.
 - b. In Supplementary Fig.9c, there are four cell populations in the plots of Dox group.
 - c. In Supplementary Fig.10, most of the plots show the same issues, including unclear grouping, many cell populations, inappropriate label position, etc.
 - d. The CD44^{high}CD62L^{low} cell gating in Supplementary Fig.11b and c is a bit too farfetched.
6. Although the authors characterized many immunological processes, these data are presented without discussion on their underlying connections and mechanisms. This should be revised in the main text.
7. The authors should consider measuring the serum levels of cytokines released from immune cells to provide further evidence on underlying mechanism.
8. The authors should measure the levels of Trp and Kyn in TME to confirm the activity of the proposed hydrogel treatment?
9. The authors should consider updating their reference. For instance, combinational immunotherapy related to Kyn-mediated immunosuppressive pathways should be cited (Nat. Commun., 2021, 12, 2934; Adv. Mater., 2021, 33, 2007247; Chem. Soc. Rev., 2020, 49, 4234-4253)
10. Minor issues:
 - a. The methodology for enzymatic activity measurement of hydrogels should be included.
 - b. Grammar errors such as "associated" instead of "associate" in Line 114, "inhibit" instead of "inhibitor" in Line 178, etc.

Reviewer #2 (Remarks to the Author): with expertise in nanomedicine

The work by Wang et. al reported a supramolecular hydrogel that was loaded with doxorubicin for

inducing ICD and kynureninase for disrupting the Kyn-mediated immunosuppressive pathways in the tumor microenvironment. This combination enabled a local chemo-immunometabolic therapy. This work is novel, I would recommend acceptance after minor revisions of the following issues:

1. In part 1 (line 108), the authors mentioned the upregulation of IDO1 and TDO2 in TNBC, but in part 5 (line 238), the authors employed a B16F10 melanoma tumor model. This design is confusing logically. Please reconsider the related content in part 1.
2. In line 63-65, the authors mentioned that small-molecule inhibitors are prone to promiscuous binding. This wording should be carefully evaluated. "Off-target" might be more appropriate.
3. In Figure 1a, what is the blackish ball referring to? Please illustrate it.
4. What's the p-values for Figs. 1d, 1e, S1a, and S1b? If the p-value is greater than the significance level, then you cannot conclude that the correlation exists even if the correlation coefficient is very high.
5. Fig. S2 is misleading. The authors did not point out whether these TCGA patients were untreated or had received treatments. Only in the former scenario would the comparison be valid.
6. What's the expression level of IDO1 and TDO2 in 4T1 and B16F10 cells, comparing to normal mouse tissues?
7. In Figs. S5 and S6, the staining of free KYNase x 3 samples appears significantly different from the others. Please explain it.
8. As Dox release from the hydrogel is essential for cellular uptake, please provide the Dox release profile and the hydrogel degradation kinetics.
9. Similarly, please also provide the KYNase release data.
10. The authors did not interpret the contour plot in Fig. 3d, which shows a highly increased ratio between CD8+ and CD4+ T cells after the combo gel treatment. What does this increase indicate in terms of immunostimulation?
11. Is there significant differences between free combo and combo gel groups, in terms of TIL and M1 macrophage abundance in secondary tumors?
12. No data is given to validate the elimination of Kyn in the TME. Please provide direct evidence to show whether hydrogel-formulated KYNase could enable sustained Kyn elimination.
13. Similarly, please also provide data to validate the induction of ICD by Dox.
14. Hyphens are missing everywhere, like "checkpoint like" in line 42, "blockade mediated" in line 45, "Kyn mediated" in line 54, and "enzyme mediated" in line 66. Please check the whole manuscript carefully to correct them.
15. There are some typos in the manuscript. For example, in line 50, "resulting from" should be "resulted from"; Line 114, "associate with" should be "associated with"; Line 137, "supermolecule interactions" should be "supermolecular interactions"; Line 178, "inhibitor" should be "inhibit". Please proofread the manuscript carefully.

Reviewer #3 (Remarks to the Author): with expertise in nanomedicine

In this article, Wang et al. co-loaded kynureninase and doxorubicin into a hydrogel formulation for chemo-immunometabolic cancer therapy. Doxorubicin is a chemotherapeutic that can induce immunogenic cell death once treated on cancer cells and kynureninase can enzymatically degrade kynurenine, a key metabolic molecule with major immunosuppression implications. Combination of these drugs can synergistically enhance tumor growth control and the slow-release enabled by the hydrogel system demonstrates major advantages with a single dose. Overall, the idea is highly novel. The research is well conducted and the results shown support the conclusions. I would recommend the acceptance of this manuscript after addressing the following concerns.

1. Can the authors elaborate more on how they measured the degradation time in Figure S3?
2. The authors should perform some drug release characterization studies to prove the advantage of slow-release using their hydrogel system. It would be interesting to see the comparison between 10% and 20% hydrogel as the authors claimed the more rigid 20% gel have slower release and is thus the mechanism of better efficacy.

3. In Figures S5 and S6b, the histological images of Free KYNase x3 appears to be significantly different from those of the other groups. Can the authors please explain this discrepancy and perhaps acquire similar looking images for a better comparison?
4. In Figure S7, the authors claimed that more apoptosis is present in mice treated with the Combo gel. However, such interpretation is difficult to draw with simply H&E stains. The authors are advised to perform TUNEL staining on these sections to identify areas and amount of apoptosis more specifically.
5. For the flow gating strategies in Figures S14-S16, the authors should include the marker on the respective axes rather than just the colors. In addition, can the authors please shift the numbers in their various flow figures? Currently the percent population are overlaying the actual populations and it's difficult to see how those populations look like. Furthermore, there are some concerns with regards to the gating strategies. In multiple instances throughout, such as Figure S9b, S10, and 3d, the authors do not gate the entire positive population. Was there a reason for this approach and if so, can the authors please elaborate? Otherwise, they are strongly encouraged to reanalyze their data with appropriate gating strategies.
6. The authors should include additional experiments demonstrating the ability of DOX to induce ICD. Examples of these can include studying calreticulin, HMGB1, and ATP expression in tumors treated with the different formulations.
7. In Figures 4a-f, the authors study the efficacy of their formulation in a bilateral tumor model. While the model is a good indicator of systemic immunity, it should not be claimed as a metastasis model. The authors are advised to change their wording as the formation of natural metastasis is incredibly complex. Alternatively, the authors can conduct additional experiments to study efficacy for metastasis through intravenously administration of cancerous cells.
8. In Figures 11b-c, the authors studied the effector memory phenotype in the spleen with CD44 and CD62L stains. However, from the flow plots provided, distinct positive populations cannot be seen. This appears to be a poor staining process as CD44 and CD62L analysis for splenocytes should be very clear. The authors should learn from other published results and repeat their study with improved staining in order to better identify the 4 different populations (CD44^{low}CD62L^{low}, CD44^{high}CD62L^{low}, CD44^{high}CD62L^{high}, and CD44^{low}CD62L^{high}).
9. It is generally accepted that B16F10 is a lowly immunogenic cell line, much more so than 4T1 (Zhong et al., BMC Genomics, 2020, 21, 2 & Lechner et al., Journal of Immunotherapy, 2013, 36 (9), 447-489). Here, the authors claimed that B16F10 is more immunogenic than 4T1. The authors are strongly advised to double check their references and claim.
10. The role of antibodies is unclear in this manuscript and as such, the inclusion of 'antibody-independent' in the title is confusing. While it is true the platform does not utilize antibodies, it also lacks many other components. As such, the authors are recommended to remove this from their title to better convey their work.

Reviewer #4 (Remarks to the Author): with expertise in cancer immunology/immunotherapy

In this manuscript, the authors report the production of a novel supramolecule hydrogel that releases doxorubicin (Dox) and kynureninase that induces tumor cell death and blocks the immunosuppressive effects of kynurenine (Kyn) metabolites when injected locally (peritumoral) into two different preclinical models. This work is of high interest to the community developing inhibitors of metabolic checkpoints such as IDO1, given the failure of IDO inhibitors in clinical trials. However, there are important points that need to be addressed to strengthen the manuscript before publication in Nature Communications:

Strengths of the work:

- 1) Innovative strategy to combine KYNase and chemotherapy
- 2) Use of two preclinical models
- 3) Only a need for a single injection of the hydrogel to induce a therapeutic effect
- 4) Good immune analysis of the tumor
- 5) Use of good controls in the experiments

6) Good experimental design with the preclinical tumor models

Major points:

- 1) Fig 1, run a similar analysis (IDO1 and TDO2 levels in melanoma as this is your second model).
- 2) Supp Fig 2, the survival analysis is quite weak and should be removed. The authors showed a higher IDO1 and TDO2 levels correlate with higher CD3e and CD8a levels in TNBC, which is fine but then they show that higher IDO1 and TDO2 levels are correlated with poor survival (no significance). These types of survival analyses are of poor nature due to many confounding factors. If you go to the same database and analyze the correlation of CD3e and CD8a with survival in TNBC, you'll see that they are associated with better survival and you are showing that both of them are correlated with higher IDO1 and TDO2 then how come the higher IDO1 and TDO2 levels are correlated with poor survival in these patients? The point is that this type of survival analysis is of low value and should not be included in the manuscript.
- 3) P.fluorescens KYNase is known to be highly immunogenic (Reference: Everett M. Stone, John Blazek, Christos Karamitros, Kendra Garrison, and George Georgiou, "Engineering and preclinical evaluation of a human enzyme immune checkpoint inhibitor for cancer therapy" in "Enzyme Engineering XXIV", Pierre Monsan, Toulouse White Biotechnology, France Magali Remaud-Simeon, LISBP-INSA, University of Toulouse, France Eds, ECI Symposium Series, (2017). https://dc.engconfintl.org/enzyme_xxiv/101). Therefore, it is important for the authors to experimentally show that the effect that they are getting is not due to the immunogenic nature of P.fluorescens KYNase.
- 4) It would be important to show the specific effect of the 20% KYNase-loaded hydrogel in comparison to the 10% hydrogel in removing Kyn in the tumor and also increase in the levels of alanine and anthranilic acid (in both tumor models).
- 5) In figure 3, the level of CD39+ CD8+ T-cells should be shown as these are considered tumor-specific T-cells.
- 6) Given the data in Figures 3e and f, it is important to show the effect of the treatment on the potential upregulation of PDL1 on treated tumors.
- 7) In figure 3, T-cell polyfunctionality should be reported similar to Fig 5. (IL2 and TNF production by TILs in ex vivo analysis).
- 8) Combination of this treatment with anti-PD1 should be explored to expand the potential use of this technology in the clinic.
- 9) In the discussion elaborate on the failure of IDO1 inhibitors in phase III clinical trials.

Minor points:

- 1) Line 21 in the abstract mentions the sequential release of Dox and KYNase but the data fails to support sequential release. This needs to be edited to concurrent release.
- 2) Line 40 a dash is missing between tumor and infiltrating immune cells
- 3) Define the acronyms in the introduction (Kyn, Dox, etc).
- 4) Line 79 dash is missing (dox- and Kyn-degrading)
- 5) Line 180, change inhibitor to inhibit
- 6) Typo on line 191 (Tregs)

REVIEWERS' COMMENTS

Reviewer #1 (Remarks to the Author): with expertise in nanomedicine

In this manuscript, the authors reported a chemo-immunometabolic strategy using a supramolecular hydrogel that releases doxorubicin (Dox) and kynureninase (KYNase) to induce immunogenic tumor cell death and disrupt Kyn-mediated immunosuppressive pathways for enhanced cancer treatment. The authors demonstrated the improved tumor inhibition and extended survival of such hydrogel in both triple negative breast cancer and melanoma. The design has a high translation potential and the data is promising. I recommend its publication after addressing the following technical issues.

We greatly appreciate the reviewer's positive feedback of the manuscript.

1. The authors demonstrated the positive correlation between IDO1/TDO2 expression and CD3E and CD8A expression in TNBC. How does this correlation vary with the tumor immunogenicity?

We have selected 8 other types of tumor cancers ranked by their immunogenicity, and performed the correlation analysis between IDO1/TDO2 expression and CD3E/CD8A expression in these tumors. Using a recently reported measure of tumor immunogenicity (Wang et al. eLife 2019;8:e49020), termed tumor immunogenicity score that not only considers tumor mutational burden but also takes the expression of the antigen processing and presenting machinery into consideration, we chose SKCM (skin cutaneous melanoma), DLBC (lymphoid neoplasm diffuse large B-cell lymphoma), COAD (colon adenocarcinoma), HNSC (head and neck squamous cell carcinoma), STAD (stomach adenocarcinoma), OV (ovarian serous cystadenocarcinoma), TGCT (testicular germ cell tumors), and LGG (brain lower grade glioma), with high to low immunogenicity. Interestingly, regardless of the immunogenicity, all tumor types analyzed showed strong positive correlation between IDO1 expression and CD3E/CD8A expression, and weak positive correlation between TDO2 expression and CD3E/CD8A expression (new **Supplementary Fig. 2**). These data indicate that upregulation of Kyn pathway may be a general mechanism for tumor immune escape.

2. The release profiles of Dox and KYNase from the hydrogel should be provided.

We have conducted drug release studies in 4T1 tumor-bearing mice. As shown in the new **Fig. 2h**, 20% hydrogel displayed more sustained release of KYNase in comparison with 10% hydrogel. 20% hydrogel also extended the release of Dox, with ~80% released in 24h, as demonstrated in the new **Supplementary Fig. 8**. The concurrent but differential release of Dox and KYNase from 20% hydrogel may potentially enhance the synergy between the two drugs.

3. According to Figure 3d and 3c, the tumor inhibition and survival improvement of the combo gel are not very effective, with the tumor rapidly growing after 20 days. Why does the treatment of Dox and KYNase (even three times) show subtle tumor inhibition effect

and similar survival with the PBS group? It is noted that the drug was intratumorally injected, which should have high tumor-killing effect.

We would like to note that all treatments were peritumorally administered, which may have smaller anti-tumor effect than intratumoral administration. We also want to point out that free combo x 3 showed statistically significant tumor inhibition ($P = 0.0130$) and survival extension ($P = 0.0035$) in comparison with the PBS group.

On the other hand, the subtle tumor inhibition effect of free combo administered once may result from the rapid clearance of both drugs when administered freely.

Regarding the efficacy of the combo gel, we note that though the tumors restarted to grow after 20 days, the growth was significantly slower than other treatment groups. The tumor regrowth may result from: 1, the exhaustion of drugs. As revealed from the drug release and TME metabolites profiles, ~80% of KYNase was released 10 days after administration, and TME Kyn level started to increase at the same time. The increase of Kyn will produce an immunosuppressive microenvironment that favors tumor growth again. We think that if the drug release from the hydrogel can be sustained longer, the tumor suppression will be further improved. 2, the upregulation of other immunosuppressive pathways that will “compensate” for the decrease of Kyn level. As revealed by the additional analysis of tumor cells and CD8⁺ TILs (new **Fig. 3h, k, Supplementary Fig. 11e, 13a**), the combo gel treatment significantly boosted PD-L1 expression on tumor cells and CD39 expression on CD8⁺ TILs, which will enhance immunosuppression through PD-1-PD-L1 axis and adenosine pathway, therefore enables tumor growth.

4. In Figure 3d, there are some issues regarding the flow cytometric plots of CD4 and CD8 cells.

a. The boundary of cell grouping is not very clear. There are multiple cell populations within the CD4⁺ gate (including PBS, Free combo, and combo gel groups). Moreover, a lot of cells were excluded from the CD8⁺ gate but seem the same population with those in the gate.

b. A group of CD4⁺CD8⁺ cells is obvious in KYNase gel and Combo gel groups, which is uncommon in flow cytometric analysis. Some explanations should be provided here.

c. It is difficult to distinguish CD4-CD8- cells in most of the groups.

d. In Dox gel group, there are many cells at the left bottom corner, which seems to be cropped from the original image.

e. The percentage labels in all flow cytometric plots cover the cell gating, which greatly affect data presentation.

We think these issues resulted from unoptimized tissue harvest, single cell preparation, staining, flow cytometry running, and data analysis. Therefore, we performed a new set of experiments to thoroughly address the reviewer’s concerns. As shown in the new **Fig. 3d**,

- a, the boundary of cell grouping is now very clear, only one population was observed for CD4⁺ and CD8⁺ gates, and all cells in the desired populations were properly gated and included in the analysis.
- b, minimal number of CD4⁺CD8⁺ cells were observed. The CD4⁺CD8⁺ populations observed before may result from unoptimized staining (nonspecific staining).
- c, CD4⁻CD8⁻ cells can now be clearly distinguished.
- d, there are no abnormal populations in all groups. We also want to note that we did not crop any images.
- e, all the labels were properly positioned to present the populations clearly.

We thank the reviewer's input on helping us to improve the manuscript.

5. Some issues of other flow cytometric analysis.

- a. *It seems that there is only one cell population in the plots of Treg analysis in Supplementary Fig.9a and CD8+IFN- γ + cell analysis in Figure 3i and Supplementary Fig.9d.*
- b. *In Supplementary Fig.9c, there are four cell populations in the plots of Dox group.*
- c. *In Supplementary Fig.10, most of the plots show the same issues, including unclear grouping, many cell populations, inappropriate label position, etc.*
- d. *The CD44^{high}CD62L^{low} cell gating in In Supplementary Fig.11b and c is a bit too farfetched.*

We greatly appreciate the reviewer's feedback and have performed a new set of experiments to thoroughly address these issues.

- a, For T_{reg} characterization, we have provided Foxp3 FMO as the control. As shown in the new **Supplementary Fig. 11f**, the two populations of T_{reg} and T_{conv} can be clearly identified. Regarding IFN γ ⁺ T cells in TdLNs, we have provided the unstimulated control for reference. As revealed in the new **Fig. 4a** and **Supplementary Fig. 14a**, after proper positioning of the labels, a small population of IFN γ ⁺ T cells can be clearly seen.
- b, As shown in the new **Supplementary Fig. 12d**, the populations of MDSCs can be clearly identified without abnormal populations.
- c, As demonstrated in the new **Supplementary Fig. 11a, 11c, 12a, 12b, 14b**, the flow cytometry profiles show clear grouping, proper gating that includes all cells of interest, no abnormal populations, and proper positioning of the labels.
- d, As revealed in the new **Supplementary Fig. 15b, c**, the CD44^{high}CD62L^{low} populations can be clearly identified.

6. Although the authors characterized many immunological processes, these data are presented without discussion on their underlying connections and mechanisms. This should be revised in the main text.

We have added more discussions on the underlying connections and mechanisms of the immunological processes in the discussion section.

7. The authors should consider measuring the serum levels of cytokines released from immune cells to provide further evidence on underlying mechanism.

We have measured the serum IFN γ and TNF α (two representative inflammatory cytokines) levels after different treatments. As shown in the new **Fig. 4d**, the combo gel treatment significantly boosted the release of these two cytokines.

8. *The authors should measure the levels of Trp and Kyn in TME to confirm the activity of the proposed hydrogel treatment?*

We have performed a longitudinal analysis of the key metabolites in TME (Trp, Kyn, and anthranilic acid) using tumor samples collected at different times after treatment with 20% and 10% hydrogels. As demonstrated in the new **Fig. 2i**, both hydrogels showed elimination of Kyn and generation of anthranilic acid 1d after treatment, with more complete Kyn elimination on day 3. The 20% hydrogel enabled more sustained depletion of Kyn (until day 10) in comparison with the 10% hydrogel (until day 5). On the other hand, the high level of Kyn and the low level of anthranilic acid (below the detection limit) of PBS-treated tumors confirmed that the elimination of Kyn resulted from KYNase treatment. All treatments didn't impact the Trp level significantly.

9. *The authors should consider updating their reference. For instance, combinational immunotherapy related to Kyn-mediated immunosuppressive pathways should be cited (Nat. Commun., 2021, 12, 2934; Adv. Mater., 2021, 33, 2007247; Chem. Soc. Rev., 2020, 49, 4234-4253)*

We have included these citations in the manuscript.

10. *Minor issues:*

a. *The methodology for enzymatic activity measurement of hydrogels should be included.*

We have added the methodology in the methods section.

b. *Grammar errors such as “associated” instead of “associate” in Line 114, “inhibit” instead of “inhibitor” in Line 178, etc.*

We have corrected these errors and thoroughly proofread the manuscript.

Reviewer #2 (Remarks to the Author): with expertise in nanomedicine

The work by Wang et. al reported a supramolecular hydrogel that was loaded with doxorubicin for inducing ICD and kynureninase for disrupting the Kyn-mediated immunosuppressive pathways in the tumor microenvironment. This combination enabled

a local chemo-immunometabolic therapy. This work is novel, I would recommend acceptance after minor revisions of the following issues:

We greatly appreciate the reviewer's favorable evaluation of our work.

1. In part 1 (line 108), the authors mentioned the upregulation of IDO1 and TDO2 in TNBC, but in part 5 (line 238), the authors employed a B16F10 melanoma tumor model. This design is confusing logically. Please reconsider the related content in part 1.

We thank the reviewer for the suggestion. We have performed similar analysis for human melanoma and included the results in part 1. We believe now the manuscript reads more logically.

2. In line 63-65, the authors mentioned that small-molecule inhibitors are prone to promiscuous binding. This wording should be carefully evaluated. "Off-target" might be more appropriate.

We have revised the wording following the reviewer's suggestion.

3. In Figure 1a, what is the blackish ball referring to? Please illustrate it.

The blackish ball represents tumor antigens released from dying tumor cells, which can be taken up by APCs in TdLNs. We have revised the fig to include the relevant legend and added the description in the manuscript.

4. What's the p-values for Figs. 1d, 1e, S1a, and S1b? If the p-value is greater than the significance level, then you cannot conclude that the correlation exists even if the correlation coefficient is very high.

We have revised these figs to include the p-values. As revealed in the new **Fig. 1d, 1e, Supplementary Fig. 1**, the p-values are smaller than the significance level.

5. Fig. S2 is misleading. The authors did not point out whether these TCGA patients were untreated or had received treatments. Only in the former scenario would the comparison be valid.

We thank the reviewer for the excellent point. We have removed the original **Supplementary Fig. 2** following both reviewer 2's and 4's suggestions.

6. What's the expression level of IDO1 and TDO2 in 4T1 and B16F10 cells, comparing to normal mouse tissues?

We couldn't find any publicly available RNA-seq datasets of 4T1 or B16F10 cells that enable us to perform such quantitative comparison. However, based on Levina, et al., *Journal of Immunology Research*, 2012, 173029, Holmgaard, et al., *J Exp Med*. 2013; 210(7): 1389–1402, and Campesato et al., *Nature Communications* 11, 4011 (2020), 4T1

cells express higher levels of IDO1 and TDO2 in comparison with normal tissues. Also, based on Holmgaard, et al., J Exp Med. 2013; 210(7): 1389–1402, Campesato et al., Nature Communications 11, 4011 (2020), Sonner et al., OncoImmunology 2016, 5, e1240858, and Spranger Sci. Transl. Med. 2013, 5, 200ra116, B16F10 cells express very low levels of IDO1 and TDO2, but they are potent inducers of IDO1 in other cells of TME.

7. In Figs. S5 and S6, the staining of free KYNase x 3 samples appears significantly different from the others. Please explain it.

We think the discrepancy resulted from H&E staining that was performed in different batches. Therefore, we performed new H&E staining for free KYNase x 3 samples and have included the new images in the new **Supplementary Fig. 5, 6**. All the images are of the same quality now.

8. As Dox release from the hydrogel is essential for cellular uptake, please provide the Dox release profile and the hydrogel degradation kinetics.

We have conducted Dox release and hydrogel degradation studies in 4T1 tumor-bearing mice. As demonstrated in the new **Supplementary Fig. 8**, 20% hydrogel extended the release of Dox, with ~80% released in 24h. Regarding to the hydrogel degradation, as shown in the new **Fig. 2g**, both 20% and 10% hydrogels displayed an initial increase in gel weight likely due to swelling, following with gradual degradation. By day 10 and 15, the 10% and 20% hydrogels had ~50% mass remaining.

9. Similarly, please also provide the KYNase release data.

We have conducted KYNase release study in 4T1 tumor-bearing mice. As shown in the new **Fig. 2h**, 20% hydrogel displayed more sustained release of KYNase in comparison with 10% hydrogel.

10. The authors did not interpret the contour plot in Fig. 3d, which shows a highly increased ratio between CD8+ and CD4+ T cells after the combo gel treatment. What does this increase indicate in terms of immunostimulation?

We have added the discussion on this fig in the manuscript. The increased ratio between CD8+ and CD4+ T cells after the combo gel treatment indicates that the treated tumor is highly inflamed, which results in infiltration of more cytotoxic T cells.

11. Is there significant differences between free combo and combo gel groups, in terms of TIL and M1 macrophage abundance in secondary tumors?

Though the combo gel treatment resulted in slightly more CD8+ and CD4+ TILs and M1-like macrophages in secondary (untreated) tumors in comparison with free combo, it did not reach statistical significance (P = 0.3195, 0.1691, and 0.7536, respectively).

12. No data is given to validate the elimination of Kyn in the TME. Please provide direct evidence to show whether hydrogel-formulated KYNase could enable sustained Kyn elimination.

We have performed a longitudinal analysis of the key metabolites in TME (Trp, Kyn, and anthranilic acid) using tumor samples collected at different times after treatment with 20% and 10% hydrogels. As demonstrated in the new **Fig. 2i**, both hydrogels showed elimination of Kyn and generation of anthranilic acid 1d after treatment, with more complete Kyn elimination on day 3. The 20% hydrogel enabled more sustained depletion of Kyn (until day 10) in comparison with the 10% hydrogel (until day 5). On the other hand, the high level of Kyn and the low level of anthranilic acid (below the detection limit) of PBS-treated tumors confirmed that the elimination of Kyn resulted from KYNase treatment. All treatments didn't impact the Trp level significantly.

13. Similarly, please also provide data to validate the induction of ICD by Dox.

We have performed FACS analysis of tumor cells to confirm ICD induction by Dox. As shown in the new **Fig. 3k and Supplementary Fig. 13b**, the combo gel treatment induced significantly increased ICD.

14. Hyphens are missing everywhere, like “checkpoint like” in line 42, “blockade mediated” in line 45, “Kyn mediated” in line 54, and “enzyme mediated” in line 66. Please check the whole manuscript carefully to correct them.

We have thoroughly checked the manuscript to correct these typos.

15. There are some typos in the manuscript. For example, in line 50, “resulting from” should be “resulted from”; Line 114, “associate with” should be “associated with”; Line 137, “supermolecule interactions” should be “supermolecular interactions”; Line 178, “inhibitor” should be “inhibit”. Please proofread the manuscript carefully.

We have thoroughly proofread the manuscript to correct these typos.

Reviewer #3 (Remarks to the Author): with expertise in nanomedicine

In this article, Wang et al. co-loaded kynureninase and doxorubicin into a hydrogel formulation for chemo-immunometabolic cancer therapy. Doxorubicin is a chemotherapeutic that can induce immunogenic cell death once treated on cancer cells and kynureninase can enzymatically degrade kynurenine, a key metabolic molecule with major immunosuppression implications. Combination of these drugs can synergistically

enhance tumor growth control and the slow-release enabled by the hydrogel system demonstrates major advantages with a single dose. Overall, the idea is highly novel. The research is well conducted and the results shown support the conclusions. I would recommend the acceptance of this manuscript after addressing the following concerns.

We sincerely appreciate the reviewer's positive evaluation of the manuscript.

1. Can the authors elaborate more on how they measured the degradation time in Figure S3?

We have added the methodology in the methods section. Briefly, 40 μ l of 5 μ M KYNase was added to 160 μ l of 1.25mM Kyn in a 96 well plate, or was first encapsulated in 50 μ l of 20% or 10% hydrogels, then added to 1.25mM Kyn in the same plate. Absorbance at 365nm was monitored with a plate reader pre-heated to 37°C, and the time needed for complete elimination of Kyn in the plate was presented for each formulation.

2. The authors should perform some drug release characterization studies to prove the advantage of slow-release using their hydrogel system. It would be interesting to see the comparison between 10% and 20% hydrogel as the authors claimed the more rigid 20% gel have slower release and is thus the mechanism of better efficacy.

We have conducted drug release studies in 4T1 tumor-bearing mice. As shown in the new **Fig. 2h**, 20% hydrogel displayed more sustained release of KYNase in comparison with 10% hydrogel. 20% hydrogel also extended the release of Dox, with ~80% released in 24h, as demonstrated in the new **Supplementary Fig. 8**. The concurrent but differential release of Dox and KYNase from 20% hydrogel may potentially enhance the synergy between the two drugs.

3. In Figures S5 and S6b, the histological images of Free KYNase x3 appears to be significantly different from those of the other groups. Can the authors please explain this discrepancy and perhaps acquire similar looking images for a better comparison?

We think the discrepancy resulted from H&E staining that was performed in different batches. Therefore, we performed new H&E staining for free KYNase x 3 samples and have included the new images in the new **Supplementary Fig. 5, 6**. All the images are of the same quality now.

4. In Figure S7, the authors claimed that more apoptosis is present in mice treated with the Combo gel. However, such interpretation is difficult to drawn with simply H&E stains. The authors are advised to perform TUNEL staining on these sections to identify areas and amount of apoptosis more specifically.

We agree with the reviewer that drawing conclusions on apoptosis from H&E staining only is difficult. Therefore, we remove the statement on apoptosis. We note in the manuscript that the combo gel treatment resulted in more disrupted tumor microstructure, which can be concluded from H&E staining.

5. For the flow gating strategies in Figures S14-S16, the authors should include the marker on the respective axes rather than just the colors. In addition, can the authors please shift the numbers in their various flow figures? Currently the percent population are overlaying the actual populations and it's difficult to see how those populations look like. Furthermore, there are some concerns with regards to the gating strategies. In multiple instances throughout, such as Figure S9b, S10, and 3d, the authors do not gate the entire positive population. Was there a reason for this approach and if so, can the authors please elaborate? Otherwise, they are strongly encouraged to reanalyze their data with appropriate gating strategies.

We greatly appreciate the reviewer's feedback and have performed a new set of experiments and optimized data analysis to thoroughly address these issues.

As shown in the new **Supplementary Fig. 18, 20, 21**, the markers together with the corresponding colors are shown on each panel.

In all flow figures, the labels are properly positioned and different populations can be clearly seen.

The analysis is performed with appropriate gating strategies now. In all flow figures, all cells in the desired populations are properly gated and included in the analysis.

6. The authors should include additional experiments demonstrating the ability of DOX to induce ICD. Examples of these can include studying calreticulin, HMGB1, and ATP expression in tumors treated with the different formulations.

We have performed FACS analysis of tumor cells for calreticulin expression to confirm ICD induction by Dox. As shown in the new **Fig. 3k and Supplementary Fig. 13b**, the combo gel treatment induced significantly increased ICD.

7. In Figures 4a-f, the authors study the efficacy of their formulation in a bilateral tumor model. While the model is a good indicator of systemic immunity, it should not be claimed as a metastasis model. The authors are advised to change their wording as the formation of natural metastasis is incredibly complex. Alternatively, the authors can conduct additional experiments to study efficacy for metastasis through intravenously administration of cancerous cells.

We thank the reviewer for the insightful comment and have revised the wording.

8. In Figures 11b-c, the authors studied the effector memory phenotype in the spleen with CD44 and CD62L stains. However, from the flow plots provided, distinct positive populations cannot be seen. This appears to be a poor staining process as CD44 and CD62L analysis for splenocytes should be very clear. The authors should learn from other published results and repeat their study with improved staining in order to better identify

the 4 different populations (CD44^{low}CD62L^{low}, CD44^{high}CD62L^{low}, CD44^{high}CD62L^{high}, and CD44^{low}CD62L^{high}).

We greatly appreciate the reviewer's input and have performed a new set of experiments to address these issues. As revealed in the new **Supplementary Fig. 15b, c**, the CD44^{high}CD62L^{low} populations (and the other 3 populations) can be clearly identified.

9. It is generally accepted that B16F10 is a lowly immunogenic cell line, much more so than 4T1 (Zhong et al., *BMC Genomics*, 2020, 21, 2 & Lechner et al., *Journal of Immunotherapy*, 2013, 36 (9), 447-489). Here, the authors claimed that B16F10 is more immunogenic than 4T1. The authors are strongly advised to double check their references and claim.

We thank the reviewer for the comment and have revised the statement in the manuscript.

10. The role of antibodies is unclear in this manuscript and as such, the inclusion of 'antibody-independent' in the title is confusing. While it is true the platform does not utilize antibodies, it also lacks many other components. As such, the authors are recommended to remove this from their title to better convey their work.

We have revised the title and the manuscript following the reviewer's suggestion.

Reviewer #4 (Remarks to the Author): with expertise in cancer immunology/immunotherapy

In this manuscript, the authors report the production of a novel supramolecule hydrogel that releases doxorubicin (Dox) and kynureninase that induces tumor cell death and blocks the immunosuppressive effects of kynurenine (Kyn) metabolites when injected locally (peritumoral) into two different preclinical models. This work is of high interest to the community developing inhibitors of metabolic checkpoints such as IDO1, given the failure of IDO inhibitors in clinical trials. However, there are important points that need to be addressed to strengthen the manuscript before publication in Nature Communications:

Strengths of the work:

- 1) Innovative strategy to combine KYNase and chemotherapy*
- 2) Use of two preclinical models*
- 3) Only a need for a single injection of the hydrogel to induce a therapeutic effect*
- 4) Good immune analysis of the tumor*

- 5) Use of good controls in the experiments
- 6) Good experimental design with the preclinical tumor models

We greatly appreciate the reviewer's favorable summary and positive feedback of our work.

Major points:

1) Fig 1, run a similar analysis (IDO1 and TDO2 levels in melanoma as this is your second model).

We thank the reviewer for the suggestion. We have performed similar analysis for human melanoma and included the results in the new **Fig. 1 and Supplementary Fig. 1**.

2) Supp Fig 2, the survival analysis is quite weak and should be removed. The authors showed a higher IDO1 and TDO2 levels correlate with higher CD3e and CD8a levels in TNBC, which is fine but then they show that higher IDO1 and TDO2 levels are correlated with poor survival (no significance). These types of survival analyses are of poor nature due to many confounding factors. If you go to the same database and analyze the correlation of CD3e and CD8a with survival in TNBC, you'll see that they are associated with better survival and you are showing that both of them are correlated with higher IDO1 and TDO2 then how come the higher IDO1 and TDO2 levels are correlated with poor survival in these patients? The point is that this type of survival analysis is of low value and should not be included in the manuscript.

We thank the reviewer for the excellent point. We have removed the original **Supplementary Fig. 2** following both reviewer 2's and 4's suggestions.

3) *P.fluorescens* KYNase is known to be highly immunogenic (Reference: Everett M. Stone, John Blazek, Christos Karamitros, Kendra Garrison, and George Georgiou, "Engineering and preclinical evaluation of a human enzyme immune checkpoint inhibitor for cancer therapy" in "Enzyme Engineering XXIV", Pierre Monsan, Toulouse White Biotechnology, France Magali Remaud-Simeon, LISBP-INSA, University of Toulouse, France Eds, ECI Symposium Series, (2017). https://dc.engconfintl.org/enzyme_xxiv/101). Therefore, it is important for the authors to experimentally show that the effect that they are getting is not due to the immunogenic nature of *P.fluorescens* KYNase.

We have performed 4T1 tumor growth study by treatment with the hydrogel loaded with heat-deactivated KYNase. Heat deactivation may change the recognition of KYNase by B cells (some epitopes may be disrupted, while some "ungenuine" epitopes may be exposed), but it should keep the recognition of KYNase by T cells intact (peptides are the same). Therefore, comparison between the therapeutic effects by KYNase-loaded hydrogel and by deactivated KYNase-loaded hydrogel should indicate whether the effects are due to immunogenicity of KYNase. As shown in the new **Supplementary Fig. 7**, KYNase-loaded hydrogel demonstrated significantly stronger anti-tumor effect in comparison with deactivated KYNase-loaded hydrogel and PBS control. Therefore, we

conclude that the therapeutic effects did not result from the immunogenicity of KYNase.

4) It would be important to show the specific effect of the 20% KYNase-loaded hydrogel in comparison to the 10% hydrogel in removing Kyn in the tumor and also increase in the levels of alanine and anthranilic acid (in both tumor models).

We have performed a longitudinal analysis of the key metabolites in TME (Trp, Kyn, and anthranilic acid) using tumor samples collected at different times after treatment with 20% and 10% hydrogels. As demonstrated in the new **Fig. 2i**, both hydrogels showed elimination of Kyn and generation of anthranilic acid 1d after treatment, with more complete Kyn elimination on day 3. The 20% hydrogel enabled more sustained depletion of Kyn (until day 10) in comparison with the 10% hydrogel (until day 5). On the other hand, the high level of Kyn and the low level of anthranilic acid (below the detection limit) of PBS-treated tumors confirmed that the elimination of Kyn resulted from KYNase treatment. All treatments didn't impact the Trp level significantly.

We would like to note that: 1, unlike anthranilic acid, alanine is not only specifically produced by the treatment. It is also a critical metabolite involved in many metabolic pathways. Therefore, its concentration is not as information-rich as that of anthranilic acid. 2, 10% hydrogel formulation was not used for B16F10 treatment, as we demonstrated in **Supplementary Fig. 4** that 20% hydrogel formulation had better anti-tumor effect. We then chose this formulation for all subsequent studies.

5) In figure 3, the level of CD39+ CD8+ T-cells should be shown as these are considered tumor-specific T-cells.

We performed a new set of experiments to characterize CD39⁺CD8⁺ T cells in TME after different treatments. As shown in the new **Fig. 3h and Supplementary Fig. 11e**, the combo gel treatment significantly increased the level of CD39⁺CD8⁺ T cells.

6) Given the data in Figures 3e and f, it is important to show the effect of the treatment on the potential upregulation of PDL1 on treated tumors.

We performed a new set of experiments to characterize PD-L1 expression on tumor cells after different treatments. As demonstrated in the new **Fig. 3k and Supplementary Fig. 13a**, the combo gel treatment significantly boosted PD-L1 expression on tumor cells, which is consistent with increased IFN γ expression in CD8⁺ and CD4⁺ TILs.

7) In figure 3, T-cell polyfunctionality should be reported similar to Fig 5. (IL2 and TNF production by TILs in ex vivo analysis).

We performed a new set of experiments to profile T cell polyfunctionality after different treatments. As shown in the new **Fig. 3f, g and Supplementary Fig. 11a-d**, the combo gel treatment significantly increased T cell polyfunctionality.

8) Combination of this treatment with anti-PD1 should be explored to expand the potential use of this technology in the clinic.

We agree with the reviewer that combination of the hydrogel with anti-PD-1 antibody will expand the potential of our platform, especially given the data of PD-L1 upregulation on treated tumors. However, we are currently investigating our platform and its combination with anti-PD-1 antibody in several other genetically engineered mouse tumor models and humanized mouse models. Therefore, we would prefer to report the data in a follow-up manuscript and we think testing this combination is beyond the scope of the current manuscript.

To indicate the potential of combining our platform with anti-PD1 antibody, we have added relevant discussions in the results and discussion sections.

9) In the discussion elaborate on the failure of IDO1 inhibitors in phase III clinical trials.

We have added the discussions on this topic.

Minor points:

1) Line 21 in the abstract mentions the sequential release of Dox and KYNase but the data fails to support sequential release. This needs to be edited to concurrent release.

We thank the reviewer for the comment. We have corrected the statement accordingly.

2) Line 40 a dash is missing between tumor and infiltrating immune cells

We have corrected the typo.

3) Define the acronyms in the introduction (Kyn, Dox, etc).

We note that these acronyms have been defined in the abstract.

4) Line 79 dash is missing (dox- and Kyn-degrading)

We have corrected the typo.

5) Line 180, change inhibitor to inhibit

We have corrected the typo.

6) Typo on line 191 (Tregs)

We have corrected the typo.

REVIEWERS' COMMENTS

Reviewer #1 (Remarks to the Author):

The authors have made their efforts to conduct additional experiments that are essential to address the reviewers questions. It has been improved and I do not have further comments.

Reviewer #2 (Remarks to the Author):

The authors have addressed all my concerns. I have no more questions so I recommend acceptance.

Reviewer #3 (Remarks to the Author):

The authors have done an excellent job in revising the manuscript. All of my prior comments have been fully addressed.

Reviewer #4 (Remarks to the Author):

I want to commend the authors for conducting a novel and rigorous study and addressing all my concerns. I would recommend publishing this manuscript in Nature Communications.

REVIEWERS' COMMENTS

Reviewer #1 (Remarks to the Author):

The authors have made their efforts to conduct additional experiments that are essential to address the reviewers questions. It has been improved and I do not have further comments.

We appreciate the reviewer's positive feedback.

Reviewer #2 (Remarks to the Author):

The authors have addressed all my concerns. I have no more questions so I recommend acceptance.

We thank the reviewer for the favorable evaluation.

Reviewer #3 (Remarks to the Author):

The authors have done an excellent job in revising the manuscript. All of my prior comments have been fully addressed.

We sincerely appreciate the reviewer's inputs on helping us to improve the manuscript.

Reviewer #4 (Remarks to the Author):

I want to commend the authors for conducting a novel and rigorous study and addressing all my concerns. I would recommend publishing this manuscript in Nature Communications.

We are grateful for the reviewer's help.